# Physics-Informed Variational State-Space Gaussian Processes

**Oliver Hamelijnck**
University of Warwick
oliver.hamelijnck@warwick.ac.uk

**Arno Solin**
Aalto University
arno.solin@aalto.fi

**Theodoros Damoulas**
University of Warwick
t.damoulas@warwick.ac.uk

## Abstract

Differential equations are important mechanistic models that are integral to many scientific and engineering applications. With the abundance of available data there has been a growing interest in data-driven physics-informed models. Gaussian processes (GPs) are particularly suited to this task as they can model complex, non-linear phenomena whilst incorporating prior knowledge and quantifying uncertainty. Current approaches have found some success but are limited as they either achieve poor computational scalings or focus only on the temporal setting. This work addresses these issues by introducing a variational spatio-temporal state-space GP that handles linear and non-linear physical constraints while achieving efficient linear-in-time computation costs. We demonstrate our methods in a range of synthetic and real-world settings and outperform the current state-of-the-art in both predictive and computational performance.

## 1 Introduction

Physical modelling is integral in modern science and engineering with applications from climate modelling [62] to options pricing [6]. Here, the key formalism to inject mechanistic physical knowledge are differential equations (DEs), which given initial and/or boundary values, are typically solved numerically [8]. In contrast machine learning is data-driven, and aims to learn latent functions from observations. However the increasing availability of data has spurred interest in combining these traditional mechanistic models with data-driven methods through physics-informed machine learning. These hybrids approaches aim to improve predictive accuracy, computational efficiency by leveraging both physical inductive biases with observations [30, 44].

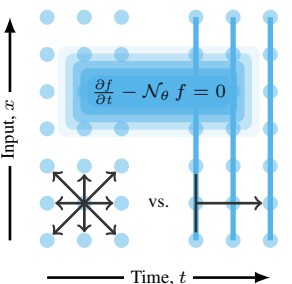

Figure 1: The state-space formalism allows for linear-time inference in the temporal dimension.

A principled way to incorporate prior physical knowledge is through Gaussian processes (GPs). GPs are stochastic processes and are a data-centric approach that facilitates the quantification of uncertainty. Recently AUTOIP was proposed in order to integrate non-linear physics into GPs [40], where solutions to ordinary and partial differential equations (ODEs, PDEs) are observed at a finite set of collocation points. This is an extension of the probabilistic meshless method (PMM, [12]) to the variational setting such that non linear equations can be incorporated. Similarly, [4] introduced HELMHOLTZ-GP, that constructs GP priors that adhere to curl and divergence-free constraints. Such properties are required for the

successful modelling of electromagnetic fields [61] and ocean currents through the Helmholtz decomposition [4]. These approaches enable the incorporation of physics but incur a cubic computational complexity from needlessly computing full covariance matrices, as illustrated in Fig. 1. For ODEs (time-series setting), extended Kalman smoothers incorporate non-linear physics (EKS) [65, 34] and recover popular ODE solvers whilst achieving linear-in-time complexity through state-space GPs [58, 25].

In this work we propose a unified physics informed state-space GP (PHYSS-GP) that is a probabilistic models where mechanistic/physics knowledge is incorporated as an inductive bias. We can handle both linear and non-linear PDEs and ODEs whilst maintaining linear-in-time computational efficiency. We additionally derive a state-space variational inference algorithm that further reduces the computational cost in the spatial dimension. We recover EKS, PMM, and HELMHOLTZ-GP as special cases, and outperform AUTOIP in terms of computational efficiency and predictive performance. In summary:

1. We derive a state-space GP that can handle spatio-temporal derivatives with a computational complexity that is linear in the temporal dimension.

2. With this we derive a unifying state-space variational inference framework that allows the incorporation of both linear and non-linear PDEs whilst achieving a linear-in-time complexity and recovering state-of-the-art methods such as EKS, PMM and HELMHOLTZ-GP.

3. We further explore three approximations, namely a structured variational posterior, spatial sparsity, and spatial minibatching, that reduce the cubic spatial computational costs to linear.

4. We showcase our methods on a variety of synthetic and real-world experiments and outperform the current state-of-the-art methods AUTOIP and HELMHOLTZ-GP both in terms computational and predictive performance.

Code to reproduce experiments is available at `https://github.com/ohamelijnck/physs_gp`.

## 2 Background on Gaussian Processes

**Gaussian processes**  A GP is a distribution on an infinite collection of random variables such that any finite subset is jointly Gaussian [50]. Given observations $\mathbf{X} \in \mathbb{R}^{N \times F}$ and $\mathbf{y} \in \mathbb{R}^N$ then

$$p(\mathbf{y}, f \,|\, \boldsymbol{\theta}) = \prod_n^N p(y_n \,|\, f(\mathbf{x}_n), \boldsymbol{\theta}) \, p(f \,|\, \boldsymbol{\theta}) \tag{1}$$

is a joint model where $p(f \,|\, \boldsymbol{\theta})$ is a zero mean GP prior with kernel $\mathbf{K}(\cdot, \cdot)$, $f(\mathbf{X}) \sim p(f(\mathbf{X}) \,|\, 0, \mathbf{K}(\mathbf{X}, \mathbf{X}))$, and $\boldsymbol{\theta}$ are (hyper) parameters. We are primarily concerned with the spatio-temporal setting where we observe $N_t$ temporal and $N_s$ spatial observations $x_{t,s} \in \mathbb{R}$, $y_{t,s} \in \mathbb{R}$ on a spatio-temporal grid. Under a Gaussian likelihood, all quantities for inference and training are available analytically and, naïvely, carry a dominant computational cost of $\mathcal{O}((N_t\, N_s)^3)$. For time series data, an efficient way to construct a GP over $f$ (and its time derivatives) is through the state-space representation of GPs. Given a Markov kernel, the temporal GP prior can be written as the solution of a discretised linear time-invariant stochastic differential equation (LTI-SDE), which at time $k$ is

$$\bar{\mathbf{f}}_{k+1} = \mathbf{A}\, \bar{\mathbf{f}}_k + q_k \quad \text{and} \quad y_k \,|\, \bar{\mathbf{f}}_k \sim p(y_k \,|\, \mathbf{H}\, \bar{\mathbf{f}}_k), \tag{2}$$

where $\mathbf{A}$ is a transition matrix, $q_k$ is Gaussian noise, $\mathbf{H}$ is an observation matrix, and $\bar{\mathbf{f}}$ is a $d$-dimensional vector of temporal derivatives $\bar{f} = [f(\cdot), \frac{\partial f(\cdot)}{\partial x}, \frac{\partial^2 f(\cdot)}{\partial x^2}, \cdots]^\top$. With appropriately designed states, matrices and densities, SDEs of this form represent a large class of GP models, and Kalman smoothing enables inference in $\mathcal{O}(N_t\, d^3)$, see [56]. In the spatio-temporal setting, when the kernel matrix decomposes as a Kronecker product $\mathbf{K} = \mathbf{K}_t \otimes \mathbf{K}_s$, then with a Markov time kernel, a state space form is admitted. This takes a particularly convenient form where the state is $\bar{\mathbf{f}}_t = [\bar{f}((\mathbf{X}_s)_1, t), \cdots, \bar{f}((\mathbf{X}_s)_{Ns}, t)]^\top$, and inference requires $\mathcal{O}(N_t(N_s\, d)^3)$, see [60].

**Derivative Gaussian processes**  One main appeal of GPs is that they are closed under linear operators. Let $\mathcal{D}[\cdot] = \mathcal{D}_t\, \mathcal{D}_s[\cdot]$ be linear functional that computes $D = d_t\, d_s$ space-time derivatives with $\mathcal{D}_t[\cdot] = \left[\cdot, \frac{\partial \cdot}{\partial t}, \frac{\partial^2 \cdot}{\partial t^2}, \cdots\right]$ and $\mathcal{D}_s[\cdot] = \left[\cdot, \frac{\partial \cdot}{\partial s}, \frac{\partial^2 \cdot}{\partial s^2}, \cdots\right]$, then at a finite set of index points, the joint prior between $\mathbf{f}$ and its time and spatial derivatives is

$$p(\bar{f}(\mathbf{X})) = \mathrm{N}\left(\mathcal{D}\mathbf{f} \,|\, \mathbf{0}, \mathcal{D}\mathbf{K}(\mathbf{X}, \mathbf{X})\mathcal{D}^*\right) \tag{3}$$

where $\bar{f}(\mathbf{X}) = \mathcal{D} f(\mathbf{X})$ and $\mathcal{D}^*$ is the adjoint of $\mathcal{D}$, meaning it operates on the second argument of the kernel [54]. When jointly modelling a single time and space derivative ($d_t = d_s = 1$) the latent functions are $\bar{\mathbf{f}} = [\mathbf{f}, \frac{\partial \mathbf{f}}{\partial s}, \frac{\partial \mathbf{f}}{\partial t}, \frac{\partial^2 \mathbf{f}}{\partial t \partial s}]^\top$ and the kernel is

$$
\bar{\mathbf{K}} = \mathcal{D} \, \mathbf{K}(\mathbf{X}, \mathbf{X}) \, \mathcal{D}^* = 
\begin{bmatrix}
\mathbf{K} & - & - & - \\
\frac{\partial}{\partial s} \mathbf{K} & \frac{\partial}{\partial s} \mathbf{K} \frac{\partial}{\partial s}^\top & - & - \\
\frac{\partial}{\partial t} \mathbf{K} & \frac{\partial}{\partial t} \mathbf{K} \frac{\partial}{\partial s}^\top & \frac{\partial}{\partial t} \mathbf{K} \frac{\partial}{\partial t}^\top & - \\
\frac{\partial^2}{\partial t \partial s} \mathbf{K} & \frac{\partial^2}{\partial t \partial s} \mathbf{K} \frac{\partial}{\partial s}^\top & \frac{\partial^2}{\partial t \partial s} \mathbf{K} \frac{\partial}{\partial t}^\top & \frac{\partial^2}{\partial t \partial s} \mathbf{K} \frac{\partial^2}{\partial t \partial s}^\top
\end{bmatrix}.
$$

This is a multi-output prior whose samples are paths of $f$ with its corresponding derivatives. This prior is commonly known as a derivative GP and has found applications in monotonic GPs [51], input-dependent noise [41, 67] and explicitly modelling derivatives [59, 17, 43]. State-space GPs can be employed in the temporal setting since the underlying state computes $f(\mathbf{x})$ with its corresponding time derivatives. In Sec. 3.1, we extend this to the spatio-temporal setting.

## 3 Physics-Informed State-Space Gaussian Processes (PHYSS-GP)

We now propose a flexible generative model for incorporating information from both data observations and (non-linear) physical mechanics. We consider general non-linear evolution equations of the form

$$
g(\mathcal{N}_\theta \, f) = \frac{\partial f}{\partial t} - \mathcal{N}_\theta \, f = 0 \tag{4}
$$

with appropriate boundary conditions, where $f : \mathbb{R}^F \to \mathbb{R}$ is the latent quantity of interest and $\mathcal{N}_\theta$ is a non-linear differential operator [49]. We assume that $g : \mathbb{R}^{P \cdot D} \to \mathbb{R}$ is measurable, and is well-defined such that there are sensible solutions to the differential equation [25]. We wish to place a GP prior over $f$ and update our beliefs after 'observing' that it should follow the solution of the differential equation. In general this is intractable and can only be handled approximately. By viewing Eqn. (4) as a loss function that measures the residual between $\frac{\partial f}{\partial t}$ and the operator $\mathcal{N}_\theta \, f$ then the right hand side (0) are virtual observations. The PDE can now be observed at a finite set of locations known as collocation points. This is a soft constraint (*i.e.* $\mathbf{f}$ is not guaranteed to follow the differential equation), but it can handle non-linear and linear mechanisms. However, there are special cases, namely curl and divergence-free constraints, that can be solved exactly. This follows from properties of vectors fields, where $f$ defines a potential function where linear combinations of its partial derivatives define vector fields that enforce these properties. To handle both of these situations we propose the following generative model

$$
\underbrace{\mathbf{F}_n = \mathbf{W} \cdot \left[ \bar{f}_q(\mathbf{X}_n) \right]^\top}_{\text{Linear Mixing}}, \quad \underbrace{\bar{f}_q \sim \mathcal{GP}(\mathbf{0}, \bar{\mathbf{K}}_q)}_{\text{Independent GP Priors}}, \tag{5}
$$

$$
\underbrace{\mathbf{y}_n^{(\mathcal{O})} = \mathbf{H}_{\mathcal{O}} \, \mathbf{F}_n + \epsilon_{\mathcal{O}}}_{\text{Data}}, \quad \underbrace{\mathbf{0}_n^{(\mathcal{C})} = g(\mathbf{F}_n)}_{\text{Collocation Points}}, \quad \underbrace{\mathbf{y}_n^{(\mathcal{B})} = \mathbf{H}_{\mathcal{B}} \, \mathbf{F}_n + \epsilon_{\mathcal{B}}}_{\text{Boundary Values}}, \tag{6}
$$

where $\bar{f}_q$ are derivative GPs (see Eqn. (3)) that are linearly mixed by $\mathbf{W} \in \mathbb{R}^{(P\,D)\times(Q\,D)}$, and $\mathbf{Y}^{(\mathcal{O})}, \mathbf{0}^{(\mathcal{C})} \in \mathbb{R}^{N \times P}$ are observations and collocation points over the P outputs and $\mathbf{Y}^{(\mathcal{B})} \in \mathbb{R}^{N \times (P\,D)}$ are boundary values over the derivatives of each output. The observation matrices $\mathbf{H}_{\mathcal{O}}, \mathbf{H}_{\mathcal{B}}$ simply select the relevant parts of $\mathbf{F}_n$. For further details on notation see App. A. In many case we want to observe the solution of the differential equation exactly, however in some cases it may be required to add observation noise $\epsilon_{\mathcal{C}}$ to the collocation points, whether for numerical reasons or to model inexact mechanics. This is a flexible generative model where different assumptions and approximations will lead to various physics informed methods such as AUTOIP, EKS, PMM, and HELMHOLTZ-GP that we will develop state space algorithms for. Additionally it is possible to learn missing physics by parameterising unknown terms in Eqn. (4) through the GP priors in Eqn. (6) (see App. B.2).

**Example 3.1** (EKS Prior and PMM)**.** We recover EKS style generative models (see Hennig et al. [25]) when the mixing weight is identity $\mathbf{W} = \mathbf{I}$, and $\epsilon_{\mathcal{C}}, \epsilon_{\mathcal{B}} \to 0$, and the non-linear transform $g$ is linearised. Let the prior be Markov $p(\bar{\mathbf{f}}) = \prod_k^{N_t} p(\bar{\mathbf{f}}_k \mid \bar{\mathbf{f}}_{k-1})$ with marginals $p(\bar{\mathbf{f}}_k) = \mathrm{N}\left( \bar{\mathbf{f}}_k \mid \mathbf{m}_k^-, \mathbf{P}_k^- \right)$. By taking a first-order Taylor linearisation $g(\bar{\mathbf{f}}_k) \simeq g(\mathbf{m}_k^-) + \frac{\partial g(\mathbf{m}_k^-)}{\partial \mathbf{m}_k^-} \delta \bar{\mathbf{f}}_k$ with

$\delta \bar{\mathbf{f}}_k \sim \mathrm{N}\left(\mathbf{0},\, \mathbf{P}_k^-\right)$ the joint is

$$p\left(\begin{bmatrix} \bar{\mathbf{f}}_k \\ \mathbf{g}_k \end{bmatrix}\right) \simeq \mathrm{N}\left(\begin{bmatrix} \bar{\mathbf{f}}_k \\ \mathbf{g}_k \end{bmatrix} \;\Big|\; \begin{bmatrix} \mathbf{m}_k^- \\ g_k(\mathbf{m}_k^-) \end{bmatrix},\; \begin{bmatrix} \mathbf{I} \\ \frac{\partial g(\mathbf{m}_k^-)}{\partial \mathbf{m}_k^-} \end{bmatrix} \mathbf{P}_k^- \begin{bmatrix} \mathbf{I} \\ \frac{\partial g(\mathbf{m}_k^-)}{\partial \mathbf{m}_k^-} \end{bmatrix}^\top \right). \tag{7}$$

This is now a form that can directly be implemented into an extended Kalman smoothing algorithm [63]. When $Q > 1$ the state $\bar{\mathbf{f}}$ is constructed by stacking the individual states of each latent [56]. With linear ODEs EKS coincides with PMM.

**Example 3.2** (HELMHOLTZ-GP and Curl and Divergence-Free Vector Fields in 2D)**.** Let $\mathbf{v} = [v_t, v_{s_1}, v_{s_2}]$ denote a 3D-vector field, then curl indicates the tendency of a vector field to rotate and divergence at a specific point indicates the tendency of the field to spread out. Curl and divergence-free fields follow

$$\nabla \times \mathbf{v} = 0 \ \ (\text{curl free}), \ \ \nabla \cdot \mathbf{v} \ = 0 \ \ (\text{div. free}) \tag{8}$$

where $\nabla = [\frac{\partial}{\partial t}, \frac{\partial}{\partial s_1}, \frac{\partial}{\partial s_2}]$. Two basic properties of vector fields state that the divergence of a curl field and the curl of a derivative field are zero [3]. Let $[f_1, f_2]$ be scalar potential functions then

$$\mathbf{v}_{\text{curl}} = \nabla f_1 \ \ (\text{curl free}), \ \ \mathbf{v}_{\text{div}} = \nabla \times \nabla f_2 \ \ (\text{div. free}) \tag{9}$$

define curl and divergence-free fields. In 2D this simplifies to using the *grad* and *rot* operators over $\mathbf{v} = [v_{s_1}, v_{s_2}]$ (see [4]). Placing GP priors over $f_q$ we incorporate this into Eqn. (6) by defining

$$\mathbf{W}_{\text{grad}} = \begin{bmatrix} 1 & 0 \\ 0 & 1 \end{bmatrix} \mathbf{H}, \ \ \mathbf{W}_{\text{rot}} = \begin{bmatrix} 0 & 1 \\ -1 & 0 \end{bmatrix} \mathbf{H} \ \text{ where } \ \mathbf{H} \ \text{ selects } \ \left[\frac{\partial f}{\partial s_1}, \frac{\partial f}{\partial s_2}\right]. \tag{10}$$

HELMHOLTZ-GP is defined as the sum of GP priors over 2D curl and divergence-free fields [4].

## 3.1 A Spatio-Temporal State-Space Prior

The generative model in Eqn. (6) contains two complications: i) it includes potential non-linearities, and ii) the independent priors are defined over latent functions with their partial derivatives which substantially increases the computational complexity. We wish to tackle both issues through state-space algorithms that are linear-in-time. We begin by deriving a state-space model that observes derivatives across space and time (see App. A.3 for the simpler time-series setting). In Sec. 3.2 we further derive a state-space variational lower bound that will enable computational speeds up in the spatial dimension.

First, we show how Kronecker structure in the kernel allows us to rewrite the model as the solution to an LTI-SDE. From the definition of $\mathcal{D}$, the separable covariance matrix has a repetitive structure that can be represented through a Kronecker product. The gram matrix is

$$\mathcal{D}\,\mathbf{K}(\mathbf{x}, \mathbf{x})\,\mathcal{D}^* = \mathbf{K}_t^{\mathcal{D}}(\mathbf{x}_t, \mathbf{x}_t) \otimes \mathbf{K}_s^{\mathcal{D}}(\mathbf{x}_s, \mathbf{x}_s) \tag{11}$$

where $\mathbf{K}_{\cdot}^{\mathcal{D}\cdot} = \begin{bmatrix} \mathbf{K}_{\cdot \cdot} & \mathbf{K}_{\cdot \cdot}\tilde{\mathcal{D}}_{\cdot}^* \\ \tilde{\mathcal{D}}_{\cdot}\mathbf{K}_{\cdot \cdot} & \tilde{\mathcal{D}}_{\cdot}\mathbf{K}_{\cdot \cdot}\tilde{\mathcal{D}}_{\cdot}^* \end{bmatrix}$ and $\tilde{\mathcal{D}}_{\cdot}[\cdot] = (\mathcal{D}_{\cdot}[\cdot])_{1:}$ excludes the underlying latent function.

To find a Kronecker form of the gram matrix over $\mathbf{X}$, we will exploit the fact that $\mathbf{X}$ is on a spatio-temporal grid and that the kernel is separable. Due to the separable structure a derivative over either the spatio (or temporal) dimension only affects the corresponding kernel, and so when considering $\mathbf{X}$, the gram matrix is still Kronecker structured:

$$\frac{\partial}{\partial s}\mathbf{K}(\mathbf{x}, \mathbf{x}) = \mathbf{K}_t(\mathbf{x}, \mathbf{x}) \cdot \frac{\partial}{\partial s}\mathbf{K}_s(\mathbf{x}, \mathbf{x}) \Rightarrow \frac{\partial}{\partial s}\mathbf{K}(\mathbf{X}, \mathbf{X}) = \mathbf{K}_t(\mathbf{X}_t, \mathbf{X}_t) \otimes \frac{\partial}{\partial s}\mathbf{K}_s(\mathbf{X}_s, \mathbf{X}_s). \tag{12}$$

The full prior over (a permuted) $\mathbf{X}$ is now given as

$$p(\bar{f}(\mathbf{X})) \cong \mathrm{N}\left(\mathbf{0},\, \mathbf{K}_t^{\mathcal{D}}(\mathbf{X}_t, \mathbf{X}_t) \otimes \mathbf{K}_s^{\mathcal{D}}(\mathbf{X}_s, \mathbf{X}_s)\right).$$

This is the form of a spatio-temporal Gaussian process with derivative kernels that can be immediately cast into a state-space form as in Eqn. (2) where $\mathbf{H} = \mathbf{I}$, as we want to observe the whole state, not just $\mathbf{f}$. The marginal likelihood and the GP posterior can now be computed using standard Kalman filtering and smoothing algorithms with a computational time of $O(N_t \cdot (N_s \cdot d_s \cdot d)^3)$. Inference in PHYSS-GP now follows Ex. 3.1 by recognising that the filtering state consists of the spatial points with there spatio-temporal derivatives. The EKS prior in Ex. 3.1 can now be simply extended to the PDE setting by placing colocation points on a spatio-temporal grid [35].

## 3.2 A State-Space Variational Lower Bound (PHYSS-VGP and PHYSS-EKS)

We now derive a variational lower bound for PHYSS-GP that maintains the computational benefits of state-space GPs. This acts as an alternative way of handling the non-linearity of $g$ in Eqn. (6), and will also enable the reduction of the cubic spatial computation complexity in Sec. 4. We start by focusing on the single latent function setting ($Q = 1$) and collect all terms that relate to observations in Eqn. (6) with $p(\mathbf{Y} \,|\, \bar{\mathbf{f}}) = \prod_n^N p(\mathbf{y}_n^{(\mathcal{O})} | \mathbf{H}_{\mathcal{O}} \, \mathbf{F}_n) \, p(\mathbf{0}_n^{(\mathcal{C})} | g(\mathbf{F}_n)) \, p(\mathbf{y}_n^{(\mathcal{B})} | \mathbf{H}_{\mathcal{B}} \, \mathbf{F}_n)$. VI frames inference as the minimisation of the Kullback–Leibler divergence between the true posterior and an approximate posterior, which leads the optimisation of the ELBO [28]:

$$\underset{q(\bar{\mathbf{f}} \,|\, \boldsymbol{\xi})}{\arg\max} \; \mathcal{L} = \mathbb{E}_{q(\bar{\mathbf{f}})} \left[ \log \frac{p(\mathbf{Y} \,|\, \bar{\mathbf{f}}) \, p(\bar{\mathbf{f}})}{q(\bar{\mathbf{f}})} \right] \tag{13}$$

where we define the approximate posterior $q(\mathbf{f} \,|\, \boldsymbol{\xi}) \triangleq \mathrm{N}(\mathbf{f} \,|\, \mathbf{m}, \, \mathbf{S})$ as a free-form Gaussian with $\boldsymbol{\xi} = (\mathbf{m}, \mathbf{S})$ and $\mathbf{m} \in \mathbb{R}^{DN \times 1}$, $\mathbf{S} \in \mathbb{R}^{DN \times DN}$. The aim is to represent the approximate posterior as a state-space GP posterior, which will enable efficient computation of the whole evidence lower bound (ELBO). We will achieve this through the use of natural gradients. The natural gradient preconditions the standard gradient with the inverse Fisher matrix, meaning the information geometry of the parameter space is taken into account, leading to faster convergence and superior performance [2, 31, 27]. For Gaussian approximate posteriors the natural gradient has a simple form [26]

$$\boldsymbol{\lambda}_k = \lambda_{k-1} + \beta \, \frac{\partial \mathcal{L}}{\partial \boldsymbol{\mu}_k} = (1 - \beta) \, \widetilde{\boldsymbol{\lambda}}_{k-1} + \beta \, \frac{\partial \mathrm{ELL}}{\partial \boldsymbol{\mu}_k} + \boldsymbol{\eta} = \widetilde{\boldsymbol{\lambda}} + \boldsymbol{\eta} \tag{14}$$

where $\boldsymbol{\lambda} = (\mathbf{S}^{-1}\mathbf{m}, 1/2\,\mathbf{S}^{-1})$ and $\boldsymbol{\mu} = (\mathbf{m}, \mathbf{m}\,\mathbf{m}^\top + \mathbf{S})$ are the natural and expectation parameterisations. This is known as conjugate variational inference (CVI) as $\widetilde{\boldsymbol{\lambda}}$ represent the natural parameters for the conjugate prior $\boldsymbol{\eta}$ [31, 10, 20, 72]. For now, we will assume that the likelihood is conjugate to ensure that $[\boldsymbol{\lambda}_k]_2$ is $p.s.d$, this will be relaxed in Sec. 5. The derivative of the ELL is

$$\frac{\partial \mathrm{ELL}}{\partial [\boldsymbol{\mu}]_2} = \sum_{\mathrm{t,s}}^{N_\mathrm{t}, N_\mathrm{s}} \frac{\partial}{\partial [\boldsymbol{\mu}]_2} \, \mathbb{E}_q \left[ \log p(\mathbf{Y}_{(\mathrm{t,s})} \,|\, \bar{\mathbf{f}}_{(\mathrm{t,s})}) \right], \tag{15}$$

where the expectation is under $q(\bar{\mathbf{f}}_{(\mathrm{t,s})})$, a $D$ dimensional Gaussian over the spatio-temporal derivatives at location $\mathbf{x}_{\mathrm{t,s}}$. Within the sum, the only elements of $[\boldsymbol{\mu}]_2$ whose gradient will propagate through the expectation are the $D \times D$ elements corresponding to these locations. These points are unique and so $\frac{\partial \mathrm{ELL}}{\partial [\boldsymbol{\mu}]_2}$ has some (permutated) block-diagonal structure, hence Eqn. (14) can be written as

$$q(\bar{\mathbf{f}}) \propto \prod_t^{Nt} \left[ \mathrm{N}(\widetilde{\mathbf{Y}}_t \,|\, \bar{\mathbf{f}}_t, \widetilde{\mathbf{V}}_t) \right] p(\bar{\mathbf{f}}) \tag{16}$$

where $\widetilde{\mathbf{Y}}_t$ is $D$-dimensional. The natural gradient update, *i.e.* $q(\bar{\mathbf{f}}_t)$ in moment parameterisation, can now be computed using Kalman smoothing in $\mathcal{O}(N_\mathrm{t} \cdot (N_\mathrm{s} \cdot d_s \cdot d)^3)$. Collecting $\widetilde{\mathbf{Y}} = \mathrm{vec}([\widetilde{\mathbf{Y}}_t])$, $\widetilde{\mathbf{V}} = \mathrm{blkdiag}\left([\widetilde{\mathbf{V}}_t]\right)$, then the ELBO can also be computed efficiently by substituting this form of $q(\bar{\mathbf{f}}_t)$ in

$$\mathcal{L} = \sum_{\mathrm{t,s}}^{N_\mathrm{t}, N_\mathrm{s}} \mathbb{E}_{q(\bar{\mathbf{f}}_{(\mathrm{t,s})})} \left[ \log p(\mathbf{Y}_{(\mathrm{t,s})} \,|\, \bar{\mathbf{f}}_{(\mathrm{t,s})}) \right] - \sum_t^{Nt} \mathbb{E}_{q(\bar{\mathbf{f}}_t)} \left[ \log \mathrm{N}(\widetilde{\mathbf{Y}}_t \,|\, \bar{\mathbf{f}}_t, \widetilde{\mathbf{V}}_t) \right] + \log p(\widetilde{\mathbf{Y}} \,|\, \widetilde{\mathbf{V}}) \tag{17}$$

where the first two terms only depend on $q(\mathcal{D}\,\mathbf{f}_t)$ and the final term is simply a by-product of running the Kalman filter, leading to a dominant computational complexity of $\mathcal{O}(N \cdot (N_\mathrm{s} \cdot d_s \cdot d)^3)$. This cost is linear in the datapoints ($N$) because the expected log likelihood above decomposes across all spatio-temporal locations. In summary we have shown that natural gradient is equivalent updating a block-diagonal likelihood that decomposes across time; hence the approximate posterior is computable via Kalman smoothing algorithms. Extending to multiple latent functions ($Q > 1$) we define a full Gaussian approximate posterior that captures all correlations between the latent functions $q(\bar{\mathbf{f}}_1, \cdots, \bar{\mathbf{f}}_Q) \triangleq \mathrm{N}\left(\bar{\mathbf{f}}_1, \cdots, \bar{\mathbf{f}}_Q \,|\, \mathbf{m}, \mathbf{S}\right)$ where $\mathbf{m} \in \mathbb{R}^{(N \times Q) \times 1}$, $\mathbf{S} \in \mathbb{R}^{(N \times Q) \times (N \times Q)}$. All the observation models in Eqn. (6) decompose across data points, hence Eqn. (16) is still block-diagonal and decomposes across time, except now each component is of dimension $Q \times N_\mathrm{t}$ as it encodes the correlations of spatial points and their spatio-temporal derivatives across the latent functions. We denote this model as PHYSS-VGP and PHYSS-EKS when using a EKS prior (see Ex. 3.1).

**Theorem 3.1.** *Let the approximate posterior be (full) Gaussian* $q(\bar{\mathbf{f}}_1, \cdots, \bar{\mathbf{f}}_Q) \triangleq \mathrm{N}\left(\bar{\mathbf{f}}_1, \cdots, \bar{\mathbf{f}}_Q \mid \mathbf{m}, \mathbf{S}\right)$ *where* $\mathbf{m} \in \mathbb{R}^{(N \times Q) \times 1}, \mathbf{S} \in \mathbb{R}^{(N \times Q) \times (N \times Q)}$. *When $g$ is linear a single natural gradient step with $\beta = 1$ recovers the optimal solution* $p(\bar{\mathbf{f}}_1, \cdots, \bar{\mathbf{f}}_Q \mid \mathbf{Y})$.

We prove this in App. A.5.4. This result not only demonstrates the optimality of our proposed inference scheme in the linear Gaussian setting, but confirms that we recover batch models like PMM and HELMHOLTZ-GP, as well as EKS (see Ex. 3.1).

# 4    Reducing the Spatial Computational Complexity

We now propose three approaches that reduce the cubic computational complexity in the number of spatial derivatives and locations. The first augments the process with inducing points that alleviate cubic costs associated with $N_s$. The second is a structured variational approximation that defines the approximate posterior only over the temporal prior and alleviates cubic costs associated with $d_s$. Finally, we introduce spatial mini-batching that alleviates linear $N_s$ costs. When used in conjunction, the dominant computation cost is $\mathcal{O}\left(N_t \cdot d_s \cdot (M_s \cdot d_t)^3\right)$. These approximations are not only useful for the state-space setting and can readily be applied to reduce the computational complexity for batch variational models (such as AUTOIP). See App. B.1 for more details.

**Spatio-Temporal Inducing Points (PHYSS-SVGP)**    In this first approximation, denoted by PHYSS-SVGP, we augment the full prior $p(\bar{\mathbf{f}})$ with inducing points. By defining these inducing points on a spatio-temporal grid, we will show that we can still exploit Markov conjugate operations through natural gradients. Let $\bar{\mathbf{u}} = \mathcal{D}\mathbf{u} \in \mathbb{R}^{M \times D}$ be inducing points at locations $\mathbf{Z} \in \mathbb{R}^{M \times F}$. From the standard SVGP formulation [27], the ELBO is

$$\mathcal{L} = \mathbb{E}_{q(\bar{\mathbf{f}}, \bar{\mathbf{u}})}\left[\log \frac{p(\mathbf{Y} \mid \bar{\mathbf{f}})\,p(\bar{\mathbf{u}})}{q(\bar{\mathbf{u}})}\right] \tag{18}$$

where $q(\bar{\mathbf{f}}, \bar{\mathbf{u}}) \triangleq p(\bar{\mathbf{f}} \mid \bar{\mathbf{u}})\,q(\bar{\mathbf{u}})$. By defining the inducing points on a spatio-temporal grid at temporal locations $\mathbf{X}_t \in \mathbb{R}_t^N$ and spatial $\mathbf{Z}_s \in \mathbb{R}^{M_s \times (F-1)}$ then the marginal $p(\bar{\mathbf{f}} \mid \bar{\mathbf{u}})$ is Gaussian with mean

$$\mu_{\mathbf{F} \mid \mathbf{U}} = \left[\mathbf{I} \otimes \mathbf{K}_s^{\mathcal{D}}(\mathbf{X}_s, \mathbf{Z}_s)\,(\mathbf{K}_s^{\mathcal{D}}(\mathbf{Z}_s, \mathbf{Z}_s))^{-1}\right] \bar{\mathbf{u}} \tag{19}$$

and variance given in Eqn. (41). This Kronecker structure allows us to again 'decouple' space and time, leading to natural gradient updates with block size $M_s \times D$, reducing the computational complexity to $\mathcal{O}(N\,(M_s \cdot d_s \cdot d)^3)$. For full details, see App. A.5.1.

**Structured Variational Inference (PHYSS-SVGP$_\mathrm{H}$)**    This second approximation, denoted as PHYSS-SVGP$_\mathrm{H}$, defines the inducing points *only* over the temporal derivatives. This is a useful approximation as it can drastically reduce the size of the filter state, making it more computationally and memory efficient. We begin by defining the joint prior as

$$p(\mathbf{F}, \mathcal{D}_t\,\mathbf{f}) = p(\mathbf{F} \mid \mathcal{D}_t\,\mathbf{f})\,p(\mathcal{D}_t\,\mathbf{f})$$

where $p(\mathbf{F} \mid \mathcal{D}_t\,\mathbf{f})$ is a Gaussian conditional with mean

$$\mathbb{E}\left[\mathbf{F} \mid \mathcal{D}_t\,\mathbf{f}\right] = \left[\mathbf{I} \otimes \widetilde{\mathbf{K}}_s^{\mathcal{D}}(\mathbf{X}_s, \mathbf{X}_s)\,\mathbf{K}_s(\mathbf{Z}_s, \mathbf{Z}_s)^{-1}\right] \mathcal{D}_t\,\mathbf{f}, \quad \text{with } \widetilde{\mathbf{K}}_s^{\mathcal{D}}(\mathbf{X}_s, \mathbf{X}_s) = \begin{bmatrix} \mathbf{K}_s(\mathbf{X}_s, \mathbf{Z}_s) \\ \mathcal{D}_s\,\mathbf{K}_s(\mathbf{X}_s, \mathbf{Z}_s) \end{bmatrix}.$$

We then define a structured variational posterior

$$q(\bar{\mathbf{f}}, \mathcal{D}_t\,\mathbf{f}) \triangleq p(\mathbf{F} \mid \mathcal{D}_t\,\mathbf{f})\,q(\mathcal{D}_t\,\mathbf{f}).$$

Substituting this into the ELBO we see that all the terms with the prior spatial derivatives cancel

$$\mathbb{E}_q\left[\log \frac{p(\mathbf{Y} \mid \bar{\mathbf{f}})\,\cancel{p(\bar{\mathbf{f}} \mid \mathcal{D}_t\,\mathbf{f})}\,p(\mathcal{D}_t\,\mathbf{f})}{\cancel{p(\mathbf{F} \mid \mathcal{D}_t\,\mathbf{f})}\,q(\mathcal{D}_t\,\mathbf{f})}\right]$$

Again, the marginal $q(\mathcal{D}_t\,\mathbf{f})$ maintains Kronecker structure, enabling Markov conjugate operations, leading to a computational cost of $\mathcal{O}(N \cdot d_s \cdot (N_s \cdot d)^3)$, see App. A.5.2. These variational approximations can simply be applied to non-state-space variational approximation, see App. B.1.

**Spatial Mini-Batching**  A standard approach for handling big data is through mini-batching where the ELL is approximated using only a data subsample [27]. Directly appling mini-batching would be of little computation benefit because computation of the ELBO requires running a Kalman smoother that iterates through all time points. Instead, we mini-batch by subsampling $B_\mathrm{s}$ spatial points

$$\mathrm{ELL} \approx \sum_\mathrm{t}^{N_\mathrm{t}} \frac{N_\mathrm{s}}{B_\mathrm{s}} \sum_i^{B_\mathrm{s}} \mathbb{E}_q \left[ \log p(\mathbf{Y}_\mathrm{t,s} \,|\, \bar{\mathbf{f}}_{\mathrm{t},i}) \right] \tag{20}$$

where $i$ is uniformly sampled. We used in conjunction with PHYSS-SVGP and PHYSS-SVGP$_\mathrm{H}$, this results in dominant costs of $\mathcal{O}\big(N_\mathrm{t}\,(M_s \cdot d_s \cdot d)^3\big)$ and $\mathcal{O}\big(N_\mathrm{t} \cdot d_s \cdot (M_s \cdot d)^3\big)$ when $B_s \ll N_\mathrm{s}$.

## 5   Handling the PSD Constraint

As discussed in Sec. 3.2 when the differential equation is non-linear, the model is no longer conjugate and the resulting natural gradients are not guaranteed to result in *p.s.d* updates. This issue has received some attention in the literature [53, 64, 39], but these approaches do not maintain an efficient conjugate representation. One distinction is [72], which uses the Gauss-Newton approximation to maintain conjugate operations. We now extend this to support spatial inducing points and non-linear transformations. Due to space we focus on PHYSS-SVGP, but see App. A.5.3 for further details. The troublesome term for the natural gradient update in Eqn. (14) is the Jacobian of the ELL *w.r.t.* to the second expectation parameter; which is not guaranteed to be *p.s.d* unless the ELL is log convex [39]. Focusing at a single location $n = (t, s)$:

$$\frac{\partial \mathrm{ELL}_n}{\partial [\boldsymbol{\mu}_k]_2} = \frac{\partial}{\partial \mathbf{S}_u} \mathbb{E}_{q(\bar{\mathbf{u}}_t)} \left[ \mathbb{E}_{p(\bar{\mathbf{f}}_n | \bar{\mathbf{u}}_t)} \left[ \log p(\mathbf{Y}_n \,|\, \bar{\mathbf{f}}_n) \right] \right]$$

we apply the Bonnet's and Price's theorem [38] to bring the differential inside the expectation and make a Gauss-Newton [19] approximation ensuring that the Jacobian is *p.s.d*

$$\frac{\partial \mathrm{ELL}}{\partial [\boldsymbol{\mu}_t]_2} \approx \sum_{n,p}^{N} \mathbb{E}_{q(\bar{\mathbf{u}}_t)} \left[ \mathbf{J}_{n,p}^\top \, \mathbf{H}_{n,p} \, \mathbf{J}_{n,p} \right], \quad \text{where } \mathbf{J}_{n,p} = \frac{\partial g_n(\mu_n)}{\partial \bar{\mathbf{u}}_t}, \quad \mathbf{H}_{n,p} = \frac{\mathrm{d}^2 \log p(\mathbf{Y}_n \,|\, g_n)}{\mathrm{d}^2 g_n}, \tag{21}$$

and $g_{n,p} = g(\bar{\mathbf{u}}_n)$ (Eqn. (4)) and $\mu_n$ is the mean of $p(\bar{\mathbf{f}}_n \,|\, \bar{\mathbf{u}}_t)$ (Eqn. (19)). When using spatial mini-batching Eqn. (21) is also subsampled.

## 6   Related Work

From the optimality of natural gradients, in the conjugate setting, we exactly recover batch GP based models such as [68, 29, 4]. Our inference scheme also applies to models that do not require derivative information  i.e. in $d_t = d_s = 1$. As a special case, we recover [20], but we have extended the inference scheme to support spatial mini-batching, allowing big spatial datasets to be used. The linear weighting matrix can be used to define a linear model of coregionalisation and its variants [7, 77, 42, 66] and through appropriately designed functionals also non-linear variants [73].

In Álvarez et al. [76] GP priors over the solution of differential equations are obtained through a stochastic forcing term but they only consider situations where the Greens function is available. In [22, 23, 57, 33], efficient state-space algorithms are derived but are limited to the temporal setting only. Similarly, Heinonen et al. [24], learn a 'free-form ODE'. In the spatio-temporal setting Krämer et al. [35] and Duffin et al. [14] (which builds [18]) derive extended Kalman filter algorithms. Additionally there are approaches to constraining GPs by linear differential equations [37, 1, 5]. More generally than [4] in [21] GP priors over the solutions to linear PDEs with constant coefficients are derived.

Beyond GP based models, physics informed neural networks (PINNs) incorporate physics by constructing a loss function between the network and the differential equation at a finite set of collocation points [48]. This amounts to a highly complex optimisation problem [36] bringing difficulties for training [70, 71] and uncertainty quantification (UQ) [16]. Current approaches to quantifying uncertainty in PINNs are based on dropout [75] and conformal predictions [47]. In recent years UQ and deep learning has received much attention however is limited by its computational cost [45].

Table 1: Test performance on the simulated damped pendulum. Time is the total wall clock time in seconds.

| MODEL | WHITEN | $C$ | TIME | RMSE | NLPD |
|---|---|---|---|---|---|
| PHYSS-GP | — | 10 | 96.33 | 0.22 | −0.09 |
| | | 100 | 112.2 | 0.05 | −0.38 |
| | | 500 | 138.98 | 0.05 | −0.72 |
| | | 1000 | 144.29 | 0.06 | −0.79 |
| AUTOIP | × | 10 | 153.52 | 0.35 | 0.41 |
| | | 100 | 195.25 | 0.2 | −0.08 |
| | | 500 | 1011.88 | 0.33 | 0.32 |
| | | 1000 | 5134.13 | 0.36 | 0.41 |
| | ✓ | 10 | 164.58 | 0.16 | −0.30 |
| | | 100 | 208.81 | 0.05 | −0.41 |
| | | 500 | 1088.31 | 0.05 | −0.75 |
| | | 1000 | 5656.62 | 0.05 | −1.39 |

Table 2: Test performance on the magnetic field strength experiment. Results are computed *w.r.t.* to the first output. Time is the average epoch time in seconds.

| | TIME | | | R SQUARED | | |
|---|---|---|---|---|---|---|
| SPATIAL SIZE | 5 | 10 | 20 | 5 | 10 | 20 |
| HELMHOLTZ-GP | 0.21 | 0.46 | 2.37 | 0.23 | 0.97 | 0.97 |
| PHYSS-GP | 0.43 | 0.60 | 1.16 | 0.21 | 0.97 | 0.97 |
| PHYSS-SVGP | 0.44 | 0.44 | 0.31 | 0.23 | 0.96 | 0.97 |
| PHYSS-VGP$_H$ | 0.29 | 0.40 | 0.35 | 0.65 | 0.86 | 0.93 |
| PHYSS-SVGP$_H$ | 0.29 | 0.28 | 0.16 | 0.65 | 0.61 | 0.74 |

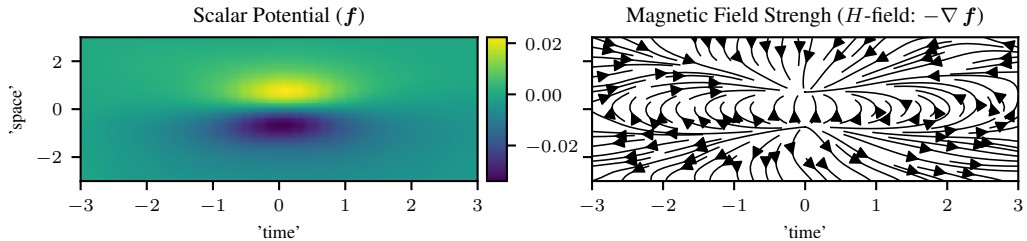

Scalar Potential ($f$)       Magnetic Field Strength ($H$-field: $-\nabla f$)

Figure 2: Curl free synthetic example. The left panel displays the learnt scalar potential functions by PHYSS-GP with $N_s = 20$, and the right panel illustrates the associated vector field.

## 7   Experiments

We now examine the performance of our PHYSS-GP methods on multiple synthetic and real-world datasets. We compare against a batch GP (no physical knowledge) and current state-of-art methods AUTOIP and HELMHOLTZ-GP. We provide more details on all experiments in App. B.

**Non-linear Damped Pendulum**   In this first synthetic example, we consider learning the non-linear dynamics of a damped swinging pendulum. This is described by a second-order differential equation

$$\frac{\mathrm{d}^2\theta}{\mathrm{d}t^2} + \sin(\theta) + b\frac{\mathrm{d}\theta}{\mathrm{dt}} = 0 \tag{22}$$

where $b > 0$. The first term is a non-linear forcing term, and the third is the damping term. With $b = 0.2$ we simulate a solution using Euler's method [9] and generate 20 points in $t \in [0, 6]$ for training and 200 in $t \in [6, 30]$ for testing, with additive Gaussian noise of variance 0.01.

We are interested in *i)* the effect of the number of collocation points, *ii)* the effect of the optimisation algorithm. To answer these questions, we compare against AUTOIP on $[10, 100, 500, 1000]$ collocation points, with and without whitening. Results are tabulated in Table 1. As expected, the predictive RMSE of all models decreases as the number of collocation points increases. Due to the cubic complexity of AUTOIP, the total time significantly increases as the number of collocation points increases. For example, when using 1000 collocation points, AUTOIP is $\approx 39$ times slower than PHYSS-GP. Interestingly, the un-whitened case performs poorly, possibly due to the nonlinearity of the differential equation making optimisation difficult. This indicates that either whitening or natural gradients are required to handle the non-linearity arising due to the differential equation.

**Curl-free Magnetic Field Strength**   In this experiment, we consider modelling the magnetic field strength of a dipole $H(r) = -\nabla\psi(r)$, where $\psi(r) = \mathbf{m}\cdot\mathbf{r}/|\mathbf{r}|^3$ is a scalar potential function [11]. Labelling the input dimensions as 'time', 'space' and '$z$', we let $\mathbf{m} = [0, 1, 0]$ and generate observations from a spatio-temporal grid with $N_t = 50$, and $N_s = [5, 10, 20]$, at $z = 1$. $H(\mathbf{r})$ is a curl-free field and so we compare the curl free part of HELMHOLTZ-GP against PHYSS-GP and its variants. HELMHOLTZ-GP and PHYSS-GP are equivalent models (as this is the conjugate setting,

Table 3: Test performance on the diffusion-reaction system. Time is the total wall clock time in seconds. PHYSS-EKS significantly outperforms all models, and due to the EKS prior only requires a 1 epoch for inference. PHYSS-SVGP_H achieves the same performance as AUTOIP but is over twice as fast.

| MODEL | RMSE | NLPD | CRPS | TIME | EPOCHS |
|---|---|---|---|---|---|
| PHYSS-EKS | **0.09** | **−1.26** | **0.038** | **$1.1 \times 10^3$** | 1 |
| PHYSS-SVGP | 0.19 | 6.80 | 0.093 | $1.4 \times 10^4$ | 20000 |
| PHYSS-SVGP_H | 0.17 | 1.69 | 0.077 | $4.8 \times 10^3$ | 20000 |
| AUTOIP | 0.17 | −0.29 | 0.065 | $1.1 \times 10^4$ | 20000 |

Theorem 3.1), and recover the same posterior and predictive distribution (up to numerical precision). However, due to the cubic-in-time complexity HELMHOLTZ-GP, at larger spatial sizes, is over 2-times slower. The hierarchical approximation is substantially faster than PHYSS-GP and performs similarly. As expected when introducing sparsity both PHYSS-SVGP and PHYSS-SVGP_H are even faster; however, this is compensated by a slight drop in predictive performance. See Fig. 2 and Table 2.

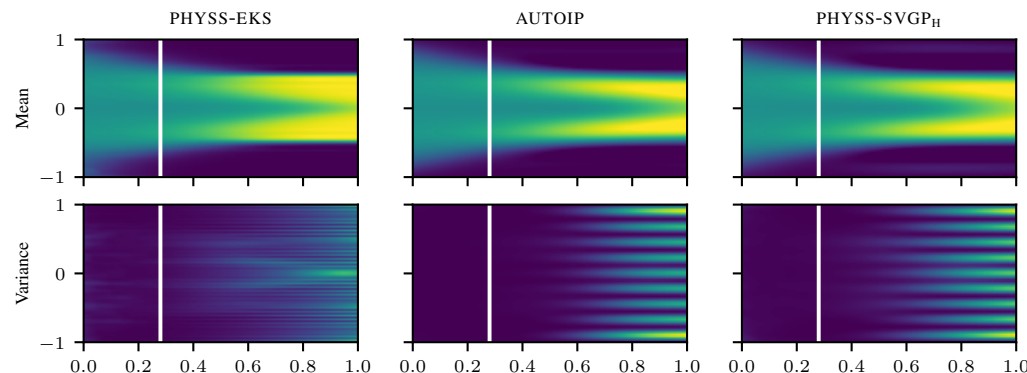

Figure 3: Results on the diffusion reaction system. The top row denotes the predictive mean, and the bottom the 95% confidence intervals. The white line denotes where the training data ends. Only PHYSS-EKS captures the sharp boundaries, due to the IWP kernel. PHYSS-SVGP_H recovers a similar solution to AUTOIP but at half the computational cost.

**Diffusion-Reaction System**   Consider a diffusion-reaction system given by an Allen-Cahn equation

$$\frac{\partial u}{\partial t} - 0.00001 \frac{\mathrm{d}^2 u}{\mathrm{d}x^2} + 5\,u^3 - 5\,u = 0 \tag{23}$$

where $x \in [-1, 1]$, $t \in [0, 1]$, $u(0, x) = x^2 \cos(\pi x)$, $u(t, -1) = u(t, 1)$ and $\frac{\partial u}{\partial x}(t, -1) = \frac{\partial u}{\partial x}(t, 1)$. Following [40], we use the solution provided by [49] and sample 256 training examples from $t \in [0, 0.28]$. We compare PHYSS-EKS (where $g$ is linerized in the EKS prior), PHYSS-SVGP and PHYSS-SVGP_H against AUTOIP. Following [40], we use a learning rate of 0.001 for Adam. For AUTOIP, we place 100 collocation points across the whole input domain on a regular grid. For both PHYSS-SVGP, and PHYSS-SVGP_H we require more collocation points in the temporal dimension and place them on a regular grid of size $20 \times 10$. For PHYSS-EKS we use an integrated Wiener kernel (IWP) on time [56] and place $100 \times 40$ collocation points. We are unable to place more collocation for AUTOIP due to computational limits. Results are presented in Fig. 3 and Table 3. PHYSS-EKS requires only a single epoch and can better handle the sharp boundaries. Our method PHYSS-SVGP_H is over twice as fast as AUTOIP whilst achieving similar predictive RMSE.

**Ocean Currents**   We now model oceanic currents in the Gulf of Mexico. We follow [4] and use the dataset provided by D'Asaro et al. [15] that has information from over $1,000$ buoys. We focus on the region in long. $[-90, -84.5]$, lat. $[26, 30]$ on 2016-02-25, by computing hourly averages. This results in $N = 42,243$ observations, and we construct a test-train split on 0.1 per cent of the data. It is infeasible to run HELMHOLTZ-GP due to data size (in Berlinghieri et al. [4], observations from only 19 buoys are used with $N = 55$). However, we run PHYSS-SVGP_H with 50 spatial inducing

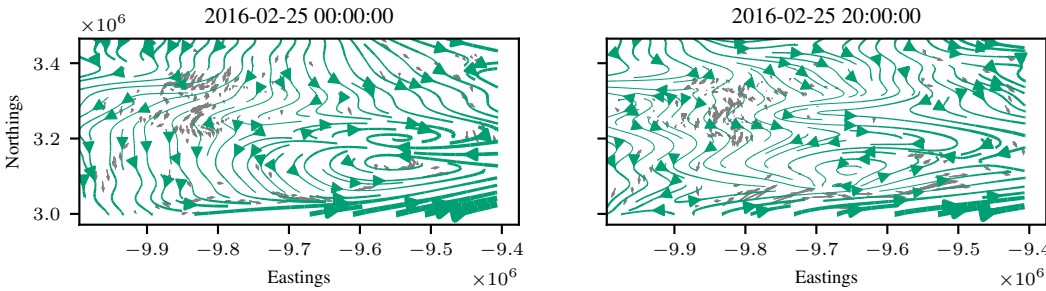

Figure 4: Predicted ocean currents by PHYSS-SVGP$_\text{H}$. True observations are in grey, and predictions in green. The thickness of the line represents uncertainty and is computed by the L2 norm of the standard deviations across both outputs.

points and a spatial mini-batch size of $10$, and plot results in Fig. 4. Our predictions are in excellent agreement with the test data, achieving an RMSE of $0.14$, NLPD of $-0.52$, CRPS of $0.078$, and an average run-time of $1.86(s)$ per epoch.

# 8   Conclusion

We introduced a physics-informed state-space GP that integrates observational data with physical knowledge. Within the variational inference framework, we derived a computationally efficient algorithm that uses Kalman smoothing to achieve linear-in-time costs. To gain further computational speed-ups, we proposed three approximations with inducing points, spatial mini batching and structured variational posteriors. When used in conjunction, they allow us to handle large-scale spatiotemporal problems. The bottleneck is always the state size, where nearest neighbours GPs [13, 74] could be explored. For highly non-linear problems, future directions could explore deep approaches [52] or more flexible kernel families [69]. One limitation is the use of the collocation method which is only enforcing the differential equation point wise, whilst future work could look at the more general methods of weighted residuals [46].

## Acknowledgments and Disclosure of Funding

OH acknowledges funding from The Alan Turing Institute PhD fellowship programme and the UKRI Turing AI Fellowship (EP/V02678X/1). AS acknowledges support from the Research Council of Finland (339730). TD acknowledges support from UKRI Turing AI Acceleration Fellowship (EP/V02678X/1) and a Turing Impact Award from the Alan Turing Institute. The authors acknowledges the University of Warwick Research Technology Platform (aquifer) for assistance in the research described in this paper. For the purpose of open access, the authors have applied a Creative Commons Attribution (CC-BY) license to any Author Accepted Manuscript version arising from this submission.

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

# Appendices

## A  Variational Approximation Derivation

### A.1  Overview of Notation

Table 4: Table of Notation

| Symbol | Size | Description |
|---|---|---|
| $N$ | – | Number of observations. |
| $Q$ | – | Number of latent functions. |
| $P$ | – | Number of latent outputs. |
| $F$ | – | Number of input features. |
| $N_\mathrm{s}$ | – | Number of spatial points. |
| $N_\mathrm{t}$ | – | Number of temporal points. |
| $d_s$ | – | Number of spatial derivatives. |
| $d_t$ | – | Number of temporal derivatives. |
| $D = d_s \cdot d_t$ | – | Total number of spatio-temporal derivatives. |
| $d$ | – | State dimension. |
| $B_\mathrm{s}$ | – | Spatial batch size. |
| $M_\mathrm{s}$ | – | Number of spatial inducing points. |
| $\mathbf{X}$ | $N \times F$ | Input data matrix. |
| $\mathbf{X}_\mathrm{s}$ | $N_\mathrm{s} \times F$ | Spatial Locations of training data. |
| $\mathbf{X}_\mathrm{t}$ | $N_\mathrm{t}$ | Temporal locations of training data. |
| $\mathbf{x}, \mathbf{X}_n, \mathbf{X}_{t,s}$ | $F$ | Single training input. |
| $\mathbf{x}_t$ | – | Temporal axis of a single training input location $\mathbf{x}$. |
| $\mathbf{x}_s$ | $F - 1$ | Spatial axes of a single training input location $\mathbf{x}$.. |
| $\mathbf{Y}$ | $N \times P$ | Output data matrix. |
| $\mathbf{Y}_n, \mathbf{Y}_{t,s}$ | $P$ | Single training output. |
| $\bar{\mathbf{f}}$ | $N_\mathrm{s} \times d$ | Filtering state. |
| $\mathbf{W}$ | $(P \times D) \times (Q \times D)$ | Mixing matrix between $Q$ latent GPs |
| $\bar{f}_q(\mathbf{X}_n)$ | $D$ | Random vector of the $D$ derivatives at location $\mathbf{X}_n$ |
| $\mathbf{F}_n$ | $(P \times D)$ | Output of linearly mixed GPs. |
| $g : \mathbb{R}^{P \cdot D} \to \mathbb{R}$ | – | Differential equation defined using $D$ spatio-temporal derivatives and $P$ outputs/states. |
| $\mathbf{Z}_\mathrm{s}$ | $\mathbb{R}^{M_s \times (F-1)}$ | Spatial Inducing Points. |
| $\mathbf{K}_s(\mathbf{X}_\mathrm{s}, \mathbf{X}_\mathrm{s})$ | $N_\mathrm{s} \times N_\mathrm{s}$ | Spatial Kernel. |
| $\mathbf{K}_t(\mathbf{X}_\mathrm{t}, \mathbf{X}_\mathrm{t})$ | $N_\mathrm{t} \times N_\mathrm{t}$ | Temporal Kernel. |
| $\mathbf{K}(\mathbf{X}, \mathbf{X})$ | $N \times N$ | Spatio-Temporal Kernel. |
| $\bar{\mathbf{K}} = \mathcal{D}\,\mathbf{K}(\mathbf{X}, \mathbf{X})\,\mathcal{D}^*$ | $(N \cdot D) \times (N \cdot D)$ | Spatio-temporal kernel over all N locations and $D$ derivatives. |
| $\mathbf{K}_t^{\mathcal{D}}(\mathbf{X}_t, \mathbf{X}_t)$ | $(N_\mathrm{t} \times d_t) \times (N_\mathrm{t} \times d_t)$ | Gram matrix over temporal derivatives. |
| $\mathbf{K}_s^{\mathcal{D}}(\mathbf{X}_t, \mathbf{X}_t)$ | $(N_\mathrm{s} \times d_s) \times (N_\mathrm{s} \times d_s)$ | Gram matrix over spatial derivatives. |

### A.2  Layout of Vectors and Matrices

We use a numerator layout for derivatives. Let $Q$ denote the number of independent latent functions and $D$ the number of derivatives computed, and let $f_{q,d}$ denote the latent GP for $d$'th derivative of the $q$'th latent function. We will need to keep track of the permutation of our data *w.r.t.* to space, time, and latent functions. Inspired by 'row-major' and 'column-major' layouts, we will use the following terminology that describes the ordering of the data across latent functions and time and space:

- **latent-data:** $\mathbf{F} = \mathbf{F}_\mathrm{ld} = [\mathbf{F}_1(\mathbf{X}), \cdots, \mathbf{F}_Q(\mathbf{X})]$ with $\mathbf{F}_q(\mathbf{X}) = [\mathbf{f}_{q,1}(\mathbf{X}), \cdots, \mathbf{f}_{q,D}(\mathbf{X})]$ which is ordered by stacking each of the latent functions on top of each other.

- **data-latent:** $\mathbf{F}_\mathrm{dl} = [\mathbf{F}_1(\mathbf{X}_n), \cdots, \mathbf{F}_Q(\mathbf{X}_n)]_n^N$ which is ordered by stacking the latent functions evaluated at each data point across all data points.

- **time-space:** $[\mathbf{f}(\mathbf{X}_\mathrm{t}^{(\mathrm{st})})]_\mathrm{t}^{N_\mathrm{t}}$ which is ordered by stacking each of the input points at each time point on top of each other. This is only applicable when there is a single latent function.

- **latent-time-space:** $[\mathbf{F}_1(\mathbf{X}_1^{(\mathrm{st})}), \cdots, \mathbf{F}_1(\mathbf{X}_{N_\mathrm{t}}^{(\mathrm{st})}), \cdots, \mathbf{F}_Q(\mathbf{X}_1^{(\mathrm{st})}), \cdots, \mathbf{F}_Q(\mathbf{X}_{N_\mathrm{t}}^{(\mathrm{st})})]$

- **time-latent-space:** $[\mathbf{F}_1(\mathbf{X}_{\mathrm{t}}^{(\mathrm{st})}), \cdots, \mathbf{F}_Q(\mathbf{X}_{\mathrm{t}}^{(\mathrm{st})})]_{\mathrm{t}}^{N_{\mathrm{t}}}$

The default order will be latent-data (and latent-time-space for spatio-temporal problems). Since all of these are just simple permutations of each other, there exists a permutation matrix that permutes between any two of the layouts above. We use the function $\pi$ to denote a function that performs this permutation such that:

$$\begin{aligned} \mathbf{F}_{\mathrm{dl}} &= \pi_{\mathrm{ld}\to\mathrm{dl}}(\mathbf{F}_{\mathrm{ld}}) \\ \mathbf{F}_{\mathrm{ld}} &= \pi_{\mathrm{dl}\to\mathrm{ld}}(\mathbf{F}_{\mathrm{dl}}) \end{aligned} \tag{24}$$

### A.3 Timeseries Setting - Single Latent Function

Let $\mathbf{X} \in \mathbb{R}^{N \times D}, \mathbf{Y} \in \mathbb{R}^{N \times P}$ be input-output observations across $P$ outputs, where $N = N_{\mathrm{t}}$. For now, we only consider the case where $Q = 1$. We assume that $f$ has a state-space representation, and we denote its state with its $D$ time derivatives as $\mathbf{F}(\mathbf{X}) = [\mathbf{f}(\mathbf{X}), \frac{\partial \mathbf{f}(\mathbf{X})}{\partial \mathbf{X}}, \frac{\partial^2 \mathbf{f}(\mathbf{X})}{\partial \mathbf{X}^2}, \cdots]$ in latent-data format. The vector $\mathbf{F}(\mathbf{X})$ is of dimension $(N \times D)$. We also use the notation $\mathbf{F}_n = \mathbf{F}(\mathbf{X}_n)$, which is a $D$-dimensional vector of the derivatives at location $\mathbf{X}_n$. The joint model is

$$\left[ \prod_n^N p(\mathbf{Y}_n \mid \mathbf{F}_n, \mathrm{DE}) \right] p(\mathbf{F}). \tag{25}$$

At this point, we place no particular restriction on the form of the likelihood, aside from decomposing across $N$. The prior $p(\mathbf{F})$ is a multivariate GP of dimension $N \times D$

$$p(\mathbf{F}) = \mathrm{N}\left( \mathbf{F} \mid \mathbf{0}, \mathcal{D}\,\mathbf{K}(\mathbf{x}, \mathbf{x})\,\mathcal{D}^* \right) \tag{26}$$

Let $q(\mathbf{F})$ be a free-form multivariate Gaussian of the same dimension as $p(\mathbf{F})$ then the corresponding ELBO is:

$$\begin{aligned} \mathcal{L} &= \mathbb{E}_{q(\mathbf{F})}\left[ \log \frac{p(\mathbf{Y} \mid \mathbf{F}, \mathrm{DE})\,p(\mathbf{F})}{q(\mathbf{F})} \right], \\ &= \mathbb{E}_{q(\mathbf{F})}\left[ \log p(\mathbf{Y} \mid \mathbf{F}, \mathrm{DE}) \right] - \mathcal{KL}\left[ q(\mathbf{F}) \,\|\, p(\mathbf{F}) \right], \\ &= \underbrace{\sum_n^N \mathbb{E}_{q(\mathbf{F}_n)}\left[ \log p(\mathbf{Y}_n \mid \mathbf{F}_n, \mathrm{DE}) \right]}_{\mathrm{ELL}} - \underbrace{\mathcal{KL}\left[ q(\mathbf{F}) \,\|\, p(\mathbf{F}) \right]}_{\mathrm{KL}}, \end{aligned} \tag{27}$$

and the marginal $q(\mathbf{F}_n)$ is a $D$-dimensional Gaussian corresponding to the $n$'th observation. The natural gradients are

$$\widetilde{\boldsymbol{\lambda}} \leftarrow (1 - \beta)\,\widetilde{\boldsymbol{\lambda}} + \beta\,\frac{\partial \mathrm{ELL}}{\partial \boldsymbol{\mu}} \Big\} \ \ \text{Surrogate likelihood update} \tag{28}$$

$$\boldsymbol{\lambda} \leftarrow \widetilde{\boldsymbol{\lambda}} + \boldsymbol{\eta} \qquad\qquad \Big\} \ \ \text{Surrogate model update} \tag{29}$$

where $\widetilde{\boldsymbol{\lambda}} = \left[ [\widetilde{\boldsymbol{\lambda}}]_1, [\widetilde{\boldsymbol{\lambda}}]_2 \right]^\top$ and $[\widetilde{\boldsymbol{\lambda}}]_1$ is an $(N \times D)$ vector and $[\widetilde{\boldsymbol{\lambda}}]_2$ an $(N \times D) \times (N \times D)$ matrix. Eqn. (29) is a sum of natural parameters, and so is the conjugate Bayesian update. Naively computing this would yield no computation speed up as the computation cost would be cubic $\mathcal{O}(N^3)$. However, the natural parameters of the likelihood $(\widetilde{\boldsymbol{\lambda}})$ are guaranteed to be block diagonal, one block per data point (if $\widetilde{\boldsymbol{\lambda}}_0$ is initialised as so). This immediately implies that Eqn. (29) can be computed using efficient Kalman filter and smoothing algorithms. The structure of $\widetilde{\boldsymbol{\lambda}}$ depends on the gradient of the expected log-likelihood $\frac{\partial \mathrm{ELL}}{\partial \boldsymbol{\mu}}$. Expanding this out

$$\frac{\partial \mathrm{ELL}}{\partial [\boldsymbol{\mu}]_2} = \sum_n^N \frac{\partial \mathbb{E}_{q(\mathbf{F}_n)}\left[ \log p(\mathbf{Y}_n \mid \mathbf{F}_n, \mathrm{DE}) \right]}{\partial [\boldsymbol{\mu}]_2} = \sum_n^N \widetilde{\boldsymbol{\mu}}_n \tag{30}$$

where each component $\widetilde{\boldsymbol{\mu}}_n$ is a $(N \times D) \times (N \times D)$ matrix that only has $D \times D$ non-zero entries; as these are the only elements that directly affect $\mathbf{F}_n$. Collecting all these submatrices into a block diagonal matrix, we have a matrix in data-latent format, however, $\frac{\partial \mathrm{ELL}}{\partial \boldsymbol{\mu}}$ is in latent-data, and so all we need to do is permute by $\mathbf{P}$:

$$\frac{\partial \mathrm{ELL}}{\partial \boldsymbol{\mu}} = \pi_{\mathrm{dl}\to\mathrm{ld}}\left( \mathrm{blkdiag}[\,\widetilde{\boldsymbol{\mu}}_1, \cdots, \widetilde{\boldsymbol{\mu}}_N\,] \right). \tag{31}$$

Converting from natural to moment paramerisation the surrogate update is:

$$q(\mathbf{F}) \propto \mathrm{N}\left(\widetilde{\mathbf{Y}} \mid \mathbf{F}, \widetilde{\mathbf{V}}\right) p(\mathbf{F})$$

$$= \left[\prod_n^N \mathrm{N}\left(\widetilde{\mathbf{Y}}_n \mid \mathbf{F}_n, \widetilde{\mathbf{V}}_n\right)\right] p(\mathbf{F}) \tag{32}$$

where $\widetilde{\mathbf{Y}}_n$ is a $D$-dimensional vector, and $\widetilde{\mathbf{V}}_n$ is a $D \times D$ matrix, and efficient Kalman filtering and smoothing algorithms can be used to compute the surrogate model update. Substituting $q(\mathbf{F})$ back into the ELBO it further simplifies:

$$\mathcal{L} = \mathbb{E}_{q(\mathbf{F})}\left[\log \frac{p(\mathbf{Y} \mid \mathbf{F}, \mathrm{DE})\,\cancel{p(\mathbf{F})}\,p(\widetilde{\mathbf{Y}} \mid \widetilde{\mathbf{V}})}{\mathrm{N}\left(\widetilde{\mathbf{Y}} \mid \mathbf{F}, \widetilde{\mathbf{V}}\right)\,\cancel{p(\mathbf{F})}}\right]$$

$$= \sum_n^N \mathbb{E}_{q(\mathbf{F}_n)}\left[\log p(\mathbf{Y}_n \mid \mathbf{F}_n, \mathrm{DE})\right] - \sum_n^N \mathbb{E}_{q(\mathbf{F}_n)}\left[\log \mathrm{N}\left(\widetilde{\mathbf{Y}}_n \mid \mathbf{F}_n, \widetilde{\mathbf{V}}_n\right)\right] + \log p(\widetilde{\mathbf{Y}} \mid \widetilde{\mathbf{V}})$$

$$\tag{33}$$

each term can be computed efficiently as the by-product of the Kalman filtering and smoothing algorithm used to compute $q(\mathbf{F})$.

### A.4  Timeseries Setting - Multiple Latent Functions

We now generalise the previous section to handle multiple independent latent functions, *i.e.* $\mathrm{Q} > 0$. The model prior now has the form

$$p(\mathbf{F}) = \prod_q^{\mathrm{Q}} p(\mathbf{F}_q) \tag{34}$$

where $p(\mathbf{F}_q)$ is a prior over $\mathbf{f}_{q,1}$ and its $D$ partial deriatives. We consider two approaches: a mean-field approximate posterior and a full Gaussian.

The first approach defined mean-field approximate posterior $q(\mathbf{F}) \triangleq \prod_q^{\mathrm{Q}} q(\mathbf{F}_q)$ where each $q(\mathbf{F}_q)$ is a free-form Gaussian of dimension $(N \times D)$. The natural gradient updates are now simply applied to each component $q(\mathbf{F}_q)$ separately, and we essentially follow the update set out in App. A.3.

The second approach is a full-Gaussian approximate posterior where $q(\mathbf{F})$ is a $(\mathrm{Q} \times D \times N)$-dimensional free-form Gaussian. In this case the ELL is

$$\mathrm{ELL} = \sum_n^N \mathbb{E}_{q(\mathbf{F}_n)}\left[\log p(\mathbf{Y}_n \mid \mathbf{F}_n, \mathrm{DE})\right] \tag{35}$$

where $q(\mathbf{F}_n)$ is of dimension $(\mathrm{Q} \times D)$. This implies that the gradient of the ELL $\frac{\partial \mathrm{ELL}}{\partial [\boldsymbol{\mu}]_2} = \sum_n^N \widetilde{\boldsymbol{\mu}}_n$ where $\widetilde{\boldsymbol{\mu}}_n$ now has $(\mathrm{Q} \times D) \times (\mathrm{Q} \times D)$ non-zero entries. Switching to moment parameterisation

$$q(\mathbf{F}) \propto \left[\prod_n^N \mathrm{N}\left(\widetilde{\mathbf{Y}}_n \mid \mathbf{F}_n, \widetilde{\mathbf{V}}_n\right)\right] p(\mathbf{F}) \tag{36}$$

where $\widetilde{\mathbf{Y}}_n$ is of dimension $(\mathrm{Q} \times D)$ and $\widetilde{\mathbf{V}}_n$ is $(\mathrm{Q} \times D) \times (\mathrm{Q} \times D)$. We can still use state-space algorithms by simply stacking the states corresponding to each $\mathbf{F}_q$ [56].

### A.5  Spatio-temporal Data - Single Latent Function

We now turn to the spatio-temporal setting where $\mathbf{X}, \mathbf{Y}$ are spatio-temporal observations on a spatio-temporal grid ordered in time-space format. We now derive the conjugate variational algorithm for PHYSS-SVGP and PHYSS-SVGP$_\mathrm{H}$. The algorithms for PHYSS-GP and PHYSS-VGP$_\mathrm{H}$ are recovered as special cases when $\mathbf{Z} = \mathbf{X}_\mathrm{s}$.

### A.5.1 Spatial Derivative Inducing Points

We follow the standard sparse variational GP procedure and augment that prior with inducing points $\mathbf{U} = \mathcal{D}\mathbf{u}$ at locations $\mathbf{Z}$. We require that the inducing points are defined on a spatial-temporal grid at the same temporal points as the data $\mathbf{X}$, such that $\mathbf{Z} = [\mathbf{Z}_t]_t^{N_t}$. This is required to ensure Kronecker structure between the inducing points and the data. The joint model is

$$p(\mathbf{Y} \mid \mathbf{F})\, p(\mathbf{F} \mid \mathbf{U})\, p(\mathbf{U}) \tag{37}$$

where

$$
\begin{aligned}
p(\mathbf{U}) &= \mathrm{N}\left(\mathbf{U} \mid \mathbf{0},\, \mathbf{K}_t^{\mathcal{P}}(\mathbf{X}_\mathrm{t}, \mathbf{X}_\mathrm{t}) \otimes \mathbf{K}_s^{\mathcal{P}}(\mathbf{Z}_\mathrm{s}, \mathbf{Z}_\mathrm{s})\right), \\
p(\mathbf{F} \mid \mathbf{U}) &= \mathrm{N}\left(\mathbf{F} \mid \mu_{\mathbf{F}\mid\mathbf{U}},\, \Sigma_{\mathbf{F}\mid\mathbf{U}}\right)
\end{aligned}
\tag{38}
$$

and the conditional mean and covariance are given by

$$
\begin{aligned}
\mu_{\mathbf{F}\mid\mathbf{U}} &= \left[\mathbf{K}_t^{\mathcal{P}}(\mathbf{X}_\mathrm{t}, \mathbf{X}_\mathrm{t}) \otimes \mathbf{K}_s^{\mathcal{P}}(\mathbf{X}_\mathrm{s}, \mathbf{Z}_\mathrm{s})\right] \left[\mathbf{K}_t^{\mathcal{P}}(\mathbf{X}_\mathrm{t}, \mathbf{X}_\mathrm{t}) \otimes \mathbf{K}_s^{\mathcal{P}}(\mathbf{Z}_\mathrm{s}, \mathbf{Z}_\mathrm{s})\right]^{-1} \mathbf{U} \\
&= \left[\mathbf{I} \otimes \mathbf{K}_s^{\mathcal{P}}(\mathbf{X}_\mathrm{s}, \mathbf{Z}_\mathrm{s})\, (\mathbf{K}_s^{\mathcal{P}}(\mathbf{Z}_\mathrm{s}, \mathbf{Z}_\mathrm{s}))^{-1}\right] \mathbf{U},
\end{aligned}
\tag{39}
$$

and

$$
\begin{aligned}
\Sigma_{\mathbf{F}\mid\mathbf{U}} =& \left[\mathbf{K}_t^{\mathcal{P}}(\mathbf{X}_\mathrm{t}, \mathbf{X}_\mathrm{t}) \otimes \mathbf{K}_s^{\mathcal{P}}(\mathbf{X}_\mathrm{s}, \mathbf{X}_\mathrm{s})\right] \\
&- \left[\mathbf{K}_{\mathbf{X},\mathbf{Z}}^{\otimes}\right] \left[\mathbf{K}_t^{\mathcal{P}}(\mathbf{X}_\mathrm{t}, \mathbf{X}_\mathrm{t}) \otimes \mathbf{K}_s^{\mathcal{P}}(\mathbf{Z}_\mathrm{s}, \mathbf{Z}_\mathrm{s})\right]^{-1} \left[\mathbf{K}_{\mathbf{X},\mathbf{Z}}^{\otimes}\right]^{\top}
\end{aligned}
\tag{40}
$$

where $\mathbf{K}_{\mathbf{X},\mathbf{Z}}^{\otimes} = \left[\mathbf{K}_t^{\mathcal{P}}(\mathbf{X}_\mathrm{t}, \mathbf{X}_\mathrm{t}) \otimes \mathbf{K}_s^{\mathcal{P}}(\mathbf{X}_\mathrm{s}, \mathbf{Z}_\mathrm{s})\right]$ which simplifies to

$$\Sigma_{\mathbf{F}\mid\mathbf{U}} = \mathbf{K}_t^{\mathcal{P}}(\mathbf{X}_\mathrm{t}, \mathbf{X}_\mathrm{t}) \otimes \left[\mathbf{K}_s^{\mathcal{P}}(\mathbf{X}_\mathrm{s}, \mathbf{X}_\mathrm{s}) - \mathbf{K}_s^{\mathcal{P}}(\mathbf{X}_\mathrm{s}, \mathbf{Z}_\mathrm{s})\, \mathbf{K}_s^{\mathcal{P}}(\mathbf{Z}_\mathrm{s}, \mathbf{Z}_\mathrm{s})^{-1}\, \mathbf{K}_s^{\mathcal{P}}(\mathbf{Z}_\mathrm{s}, \mathbf{X}_\mathrm{s})\right] \tag{41}$$

Due to the Kronecker structure, the marginal at time $t$ only depends on the inducing points in that time slice so we can still get a CVI-style update that can be computed using a state-space model. To see why we again look at the Jacobian of the ELL: $\frac{\partial \mathrm{ELL}}{\partial [\boldsymbol{\mu}]_2} = \sum_n^N \widetilde{\boldsymbol{\mu}}_n$ where $\widetilde{\boldsymbol{\mu}}_n$ now has $(D \times M) \times (D \times M)$ non-zero entries, which corresponding to needed all $M$ spatial inducing points with there derivatives to predict at a single time point. This is similar to the time series setting, except we have now predicted in space to compute marginals of $q(\mathbf{F})$. To be complete, we write that the marginal $q(\mathbf{U})$ is

$$q(\mathbf{U}) \propto \left[\prod_t^{N_t} \mathrm{N}\left(\widetilde{\mathbf{Y}}_t \mid \mathbf{F}_t, \widetilde{\mathbf{V}}_t\right)\right] p(\mathbf{F}) \tag{42}$$

where $\widetilde{\mathbf{Y}}_t$ and $\mathbf{F}_t$ are vectors of dimension $(D \times M)$, and $\widetilde{\mathbf{V}}_t$ is a matrix of dimension $(D \times M) \times (D \times M)$. The marginals $q(\mathbf{U}_t)$, and the corresponding marginal likelihood $p(\widetilde{\mathbf{Y}} \mid \widetilde{\mathbf{V}})$ can be computed by running a Kalman filter and smoother in $\mathcal{O}(N_t \cdot (M_s \cdot d_s \cdot d)^3)$. The marginal $q(\mathbf{F}) = \mathrm{N}\left(\mathbf{F} \mid \mu_{\mathbf{F}}, \Sigma_{\mathbf{F}}\right)$ where

$$\mu_{\mathbf{F}} = \left[\mathbf{I} \otimes \mathbf{K}_s^{\mathcal{P}}(\mathbf{X}_\mathrm{s}, \mathbf{Z}_\mathrm{s})\, (\mathbf{K}_s^{\mathcal{P}}(\mathbf{Z}_\mathrm{s}, \mathbf{Z}_\mathrm{s}))^{-1}\right] \mathbf{m} \tag{43}$$

and

$$\Sigma_{\mathbf{F}} = \Sigma_{\mathbf{F}\mid\mathbf{U}} + \left[\mathbf{I} \otimes \mathbf{K}_s^{\mathcal{P}}(\mathbf{X}_\mathrm{s}, \mathbf{Z}_\mathrm{s})\, (\mathbf{K}_s^{\mathcal{P}}(\mathbf{Z}_\mathrm{s}, \mathbf{Z}_\mathrm{s}))^{-1}\right] \mathbf{S} \left[\mathbf{I} \otimes \mathbf{K}_s^{\mathcal{P}}(\mathbf{X}_\mathrm{s}, \mathbf{Z}_\mathrm{s})\, (\mathbf{K}_s^{\mathcal{P}}(\mathbf{Z}_\mathrm{s}, \mathbf{Z}_\mathrm{s}))^{-1}\right]^{\top} \tag{44}$$

### A.5.2 Structured Approximate Posterior With Spatial Inducing Points

We now derive the algorithm for the case of the structured approximate posterior with spatial inducing points. The key is to define the free-form approximate posterior over the inducing points and their temporal derivatives and then use the model conditional to compute the spatial derivatives. The model is

$$p(\mathbf{Y} \mid \mathbf{F})\, p(\mathbf{F} \mid \mathcal{D}_\mathrm{t}\mathbf{u})\, p(\mathcal{D}_\mathrm{t}\mathbf{u}). \tag{45}$$

Each term is

$$
\begin{aligned}
p(\mathcal{D}_\mathrm{t}\mathbf{u}) &= \mathrm{N}\left(\mathcal{D}_\mathrm{t}\mathbf{u} \mid \mathbf{0},\, \mathcal{D}_\mathrm{t}\mathbf{K}(\mathbf{Z}, \mathbf{Z})\mathcal{D}_\mathrm{t}^{*}\right), \\
p(\mathbf{F} \mid \mathcal{D}_\mathrm{t}\mathbf{u}) &= \mathrm{N}\left(\mathbf{F} \mid \mu_{\mathbf{F}\mid\mathbf{U}_t},\, \Sigma_{\mathbf{F}\mid\mathbf{U}_t}\right)
\end{aligned}
\tag{46}
$$

where

$$
\begin{aligned}
\mu_{\mathbf{F}\mid\mathbf{U}_t} &= \left[\mathbf{K}_t^{\mathcal{P}}(\mathbf{X}_\mathrm{t}, \mathbf{X}_\mathrm{t}) \otimes \widetilde{\mathbf{K}}_s^{\mathcal{P}}(\mathbf{X}_\mathrm{s}, \mathbf{X}_\mathrm{s})\right] \left[\mathbf{K}_t^{\mathcal{P}}(\mathbf{X}_\mathrm{t}, \mathbf{X}_\mathrm{t}) \otimes \mathbf{K}_s(\mathbf{Z}_\mathrm{s}, \mathbf{Z}_\mathrm{s})\right]^{-1} \mathcal{D}_\mathrm{t}\mathbf{u} \\
&= \left[\mathbf{I} \otimes \widetilde{\mathbf{K}}_s^{\mathcal{P}}(\mathbf{X}_\mathrm{s}, \mathbf{X}_\mathrm{s})\, \mathbf{K}_s(\mathbf{Z}_s, \mathbf{Z}_s)^{-1}\right] \mathcal{D}_\mathrm{t}\mathbf{u}
\end{aligned}
\tag{47}
$$

and

$$\Sigma_{\mathbf{F} \mid \mathbf{U}_t} = \left[ \mathbf{K}_t^{\mathcal{D}}(\mathbf{X}_t, \mathbf{X}_t) \otimes \mathbf{K}_s^{\mathcal{D}}(\mathbf{X}_s, \mathbf{X}_s) \right]$$
$$- \left[ \widetilde{\mathbf{K}}_{\mathbf{X}, \mathbf{Z}_s}^{\otimes} \right] \left[ \mathbf{K}_t^{\mathcal{D}}(\mathbf{X}_t, \mathbf{X}_t) \otimes \mathbf{K}_s(\mathbf{Z}_s, \mathbf{Z}_s) \right]^{-1} \left[ \widetilde{\mathbf{K}}_{\mathbf{X}, \mathbf{Z}_s}^{\otimes} \right]^{\top} \tag{48}$$

where $\widetilde{\mathbf{K}}_{\mathbf{X}, \mathbf{Z}_s}^{\otimes} = \mathbf{K}_t^{\mathcal{D}}(\mathbf{X}_t, \mathbf{X}_t) \otimes \widetilde{\mathbf{K}}_s^{\mathcal{D}}(\mathbf{X}_s, \mathbf{Z}_{ss})$ and

$$\widetilde{\mathbf{K}}_s^{\mathcal{D}}(\mathbf{X}_s, \mathbf{X}_s) = \begin{bmatrix} \mathbf{K}_s(\mathbf{X}_s, \mathbf{Z}_s) \\ \mathcal{D}_s \, \mathbf{K}_s(\mathbf{X}_s, \mathbf{Z}_s) \end{bmatrix}. \tag{49}$$

The approximate posterior is defined as

$$q(\mathbf{F}, \mathcal{D}_t \, \mathbf{u}) = p(\mathbf{F} \mid \mathcal{D}_t \, \mathbf{u}) \, q(\mathcal{D}_t \, \mathbf{u}) \tag{50}$$

where $q(\mathcal{D}_t \, \mathbf{u}$ is a free-form Gaussian of dimension $(Nd \times N_t \times M_s)$. The rest of the derivation simply follows App. A.5.1 by simpling substituting $\widetilde{\mathbf{K}}_s^{\mathcal{D}}(\mathbf{X}_s, \mathbf{Z}_{ss})$ into the corresponding conditionals. The final result is that the approximate posterior decomposes as

$$q(\mathbf{U}) \propto \left[ \prod_t^{N_t} \mathrm{N}\left( \widetilde{\mathbf{Y}}_t \mid \mathbf{F}_t, \widetilde{\mathbf{V}}_t \right) \right] p(\mathbf{F}) \tag{51}$$

where $\widetilde{\mathbf{Y}}_t$ and $\mathbf{F}_t$ are vectors of dimension $(d_t \times M_s)$, and $\widetilde{\mathbf{V}}_t$ is a matrix of dimension $(d_t \times M_s) \times (d_t \times M_s)$. The marginals $q(\mathbf{U}_t)$, and the corresponding marginal likelihood $p(\widetilde{\mathbf{Y}} \mid \widetilde{\mathbf{V}})$ can be computed by running a Kalman filter and smoother in $\mathcal{O}(N_t \cdot (M_s \cdot d)^3)$, which compared to App. A.5.1 is *not* cubic in the number of spatial derivatives.

### A.5.3 Gauss-Newton Natural Gradient Approximation

We now provide the full derivation of the Gauss-Newton approximation of the natural gradient used to ensure *p.s.d* updates. We will make use of the following identities, known as the Bonnet and Price theorems (see, [38]),

$$\frac{\partial}{\partial \mu} \mathbb{E}_{q(\mathbf{f} \mid \mu, \Sigma)} \left[ \ell(\mathbf{f}) \right] = \mathbb{E}_{q(\mathbf{f} \mid \mu, \Sigma)} \left[ \frac{\partial}{\partial \mathbf{f}} \ell(\mathbf{f}) \right] \tag{52}$$

$$\frac{\partial}{\partial \Sigma} \mathbb{E}_{q(\mathbf{f} \mid \mu, \Sigma)} \left[ \ell(\mathbf{f}) \right] = \frac{1}{2} \mathbb{E}_{q(\mathbf{f} \mid \mu, \Sigma)} \left[ \frac{\partial^2}{\partial \mathbf{f} \, \partial \mathbf{f}^{\top}} \ell(\mathbf{f}) \right] \tag{53}$$

which describes how to bring derivatives inside expectations. To ease notations, we work with a more general description of the model presented in the main paper, where we have multiple independent latent functions and use $T_p$ to denote likelihood-specific functions which, for example, can be used to represent DE or as the identity of standard Gaussian likelihoods. The model is

$$p(\mathbf{u}_q) = \mathrm{N}\left( \mathbf{u}_q \mid 0, \mathbf{K}_q \right)$$
$$p(\mathbf{f}_q \mid \mathbf{u}_q) = \mathrm{N}\left( \mathbf{f}_q \mid \mu_{\mathbf{f}_q \mid \mathbf{u}_q}, \Sigma_{\mathbf{f}_q \mid \mathbf{u}_q} \right) \tag{54}$$
$$\mathbf{Y}_{n,q} = p(\mathbf{Y}_{n,q} \mid T_p(\mathbf{f}_{n,1}, \ldots, \mathbf{f}_{n,Q}))$$

where the shapes are $\mathbf{u}_q \in \mathbb{R}^M$, $\mathbf{f}_q \in \mathbb{R}^N$, $T_p : \mathbb{R}^Q \to \mathbb{R}^P$, $\mathbf{Y} \in \mathbb{R}^{N \times P}$, and $\mathbf{Y}_{n,p} \in \mathbb{R}$. The variational approximation is

$$q(\mathbf{U}) = \mathrm{N}\left( \mathbf{U} \mid \mathbf{m}, \mathbf{S} \right) \tag{55}$$

where $\mathbf{U} = [\mathbf{u}_1, \cdots, \mathbf{u}_Q]$, $\mathbf{m} \in \mathbb{R}^{QM \times 1}$ and $\mathbf{S} \in \mathbb{R}^{QM \times QM}$. Let $\mathbf{F} = [\mathbf{f}_1, \ldots, \mathbf{f}_Q]$. The expected log-likelihood of the variational approximation is

$$\mathrm{ELL} = \mathbb{E}_{q(\mathbf{U})} \left[ \mathbb{E}_{p(\mathbf{F} \mid \mathbf{U})} \left[ \sum_{n,p} \log p(\mathbf{Y}_{n,p} \mid T_p(\mathbf{F}_{n,p})) \right] \right]$$

$$= \sum_{n,p} \mathbb{E}_{q(\mathbf{U})} \left[ \mathbb{E}_{p(\mathbf{F}_n \mid \mathbf{U})} \left[ \log p(\mathbf{Y}_{n,p} \mid T_p(\mathbf{F}_{n,p})) \right] \right] \tag{56}$$

$$= \sum_{n,p} \mathbb{E}_{q(\mathbf{U}_k)} \left[ \mathbb{E}_{p(\mathbf{F}_n \mid \mathbf{U}_k)} \left[ \log p(\mathbf{Y}_{n,p} \mid T_p(\mathbf{F}_{n,p})) \right] \right]$$

where $k = t(n)$ is the time period associated with data $\mathbf{X}_n$. Here $\mathbf{U}_k$ are the spatial inducing points at time $t(n)$ and hence $\mathbf{U}_k \in \mathbb{R}^{QM_s}$. We need to compute

$$
\begin{aligned}
\frac{\partial \text{ELL}}{\partial \mathbf{S}} &= \sum_{n,p} \frac{\partial}{\partial \mathbf{S}} \, \mathbb{E}_{q(\mathbf{U}_k)} \big[ \, \mathbb{E}_{p(\mathbf{F}_n \mid \mathbf{U}_k)} \big[ \log p(\mathbf{Y}_{n,p} \mid T_p(\mathbf{F}_{n,p})) \big] \big] \\
&= \sum_{n,p} \mathbf{P}_k \cdot \frac{\partial}{\partial \mathbf{S}_k} \, \mathbb{E}_{q(\mathbf{U}_k)} \big[ \, \mathbb{E}_{p(\mathbf{F}_n \mid \mathbf{U}_k)} \big[ \log p(\mathbf{Y}_{n,p} \mid T_p(\mathbf{F}_{n,p})) \big] \big] \cdot \mathbf{P}_k^\top \\
&= \sum_{n,p} \mathbf{P}_k \cdot \mathbb{E}_{q(\mathbf{U}_k)} \left[ \frac{\partial^2}{\partial \mathbf{U}_k \, \partial \mathbf{U}_k^\top} \, \mathbb{E}_{p(\mathbf{F}_n \mid \mathbf{U}_k)} \big[ \log p(\mathbf{Y}_{n,p} \mid T_p(\mathbf{F}_{n,p})) \big] \right] \cdot \mathbf{P}_k^\top \\
&\overset{\text{delta}}{\approx} \sum_{n,p} \mathbf{P}_k \cdot \mathbb{E}_{q(\mathbf{U}_k)} \left[ \frac{\partial^2}{\partial \mathbf{U}_k \, \partial \mathbf{U}_k^\top} \, \log p(\mathbf{Y}_{n,p} \mid T_p(\mathbf{F}_{n,p}^*)) \right] \cdot \mathbf{P}_k^\top \\
&\overset{\text{Gauss-Newton}}{\approx} \sum_{n,p} \mathbf{P}_k \cdot \mathbb{E}_{q(\mathbf{U}_k)} \left[ \left[ \frac{\partial T_p(\mathbf{F}_{n,p}^*)}{\partial \mathbf{U}_k} \right]^\top \frac{\partial^2 \log p(\mathbf{Y}_{n,p} \mid T_p)}{\partial T_p \, \partial T_p^\top} \left[ \frac{\partial T_p(\mathbf{F}_{n,p}^*)}{\partial \mathbf{U}_k} \right] \right] \cdot \mathbf{P}_k^\top \\
&= \sum_{n,p} \mathbf{P}_k \cdot \mathbb{E}_{q(\mathbf{U}_k)} \big[ \mathbf{J}_k^\top \mathbf{H}_k \mathbf{J} \big] \cdot \mathbf{P}_k^\top
\end{aligned}
$$

(57)

where the shapes are

$$
\mathbf{J} \in QM_s \times 1 \tag{58}
$$
$$
\mathbf{H}_k \in 1 \times 1 \tag{59}
$$

and $\mathbf{P}_k$ is a permutation matrix that permutes from data-latent to latent-data format. In implementation, we do not need to perform this permutation as we only require the blocks $\frac{\partial \text{ELL}}{\partial \mathbf{S}_k}$ but write it here for completeness.

### A.5.4 Optimality of Natural Gradients In Linear Models

**Theorem A.1.** *Consider a linear multi-task model of the form*

$$
\begin{aligned}
f_q(\cdot) &\sim \mathcal{GP}(0, \mathbf{K}_q) \\
\widetilde{\mathbf{f}}(\mathbf{x}) &= [f_q(\mathbf{x})]_{q=1}^{Q} \\
\mathbf{Y}_n &= \mathbf{W} \widetilde{\mathbf{f}}(\mathbf{X}_n) + \psi \ \text{where} \ \psi \sim \mathrm{N}\left( \mathbf{0}, \, \text{blkdiag}\left( [\sigma_p^2]_{p=1}^{P} \right) \right)
\end{aligned}
\tag{60}
$$

*then under a full Gaussian variational approximate posterior*

$$
q(\widetilde{\mathbf{f}}) \triangleq \mathrm{N}\left( \widetilde{\mathbf{f}} \mid \mathbf{m}, \, \mathbf{S} \right) \tag{61}
$$

*where $\mathbf{m} \in \mathbb{R}^{(N \times Q) \times 1}, \mathbf{S} \in \mathbb{R}^{(N \times Q) \times (N \times Q)}$ then the natural gradient update with a learning rate of 1 recovers the optimal solution $p(\widetilde{\mathbf{f}} \mid \mathbf{Y})$.*

*Proof.* To prove this we first derive the natural parameters of the posterior $p(\widetilde{\mathbf{f}} \mid \mathbf{Y})$. We then derive the closed form expression of the natural parameter update and show that they recover that of the posterior. Let

$$
\mathbf{F}(\mathbf{X}) = \widetilde{\mathbf{W}} \widetilde{\mathbf{f}}(\mathbf{X}) \sim \mathrm{N}\left( \mathbf{F}(\mathbf{X}) \mid \mathbf{0}, \, \widetilde{\mathbf{W}} \widetilde{\mathbf{K}}_{\mathbf{X},\mathbf{X}} \widetilde{\mathbf{W}}^\top \right) \tag{62}
$$

where $\widetilde{\mathbf{K}}_{\mathbf{X},\mathbf{X}} = \text{blkdiag}\left( [\mathbf{K}_q]_{q=1}^{Q} \right)$ and $\widetilde{\mathbf{W}} = \mathbf{W} \otimes \boldsymbol{I}$ then the posterior $p(\mathbf{F}(\mathbf{X}) \mid \mathbf{Y})$ is Gaussian

$$
p(\mathbf{F}(\mathbf{X}) \mid \mathbf{Y}) = \mathrm{N}\left( \mathbf{F}(\mathbf{X}) \mid \mu_{\mathbf{F} \mid \mathbf{Y}}, \, \Sigma_{\mathbf{F} \mid \mathbf{Y}} \right) \tag{63}
$$

with

$$
\begin{aligned}
\mu_{\mathbf{F} \mid \mathbf{Y}} &= \left[ \widetilde{\mathbf{W}} \widetilde{\mathbf{K}}_{\mathbf{X},\mathbf{X}} \widetilde{\mathbf{W}}^\top \right] \left[ \widetilde{\mathbf{W}} \widetilde{\mathbf{K}}_{\mathbf{X},\mathbf{X}} \widetilde{\mathbf{W}}^\top + \Phi \right]^{-1} \mathbf{Y} \\
\Sigma_{\mathbf{F} \mid \mathbf{Y}} &= \left[ \left[ \widetilde{\mathbf{W}} \widetilde{\mathbf{K}}_{\mathbf{X},\mathbf{X}} \widetilde{\mathbf{W}}^\top \right]^{-1} + \Phi^{-1} \right]^{-1}.
\end{aligned}
\tag{64}
$$

The covariance matrix can be simplified by invoking Woodbury's identity twice

$$
\begin{aligned}
\Sigma_{\mathbf{F}\,|\,\mathbf{Y}} &= \left[\widetilde{\mathbf{W}}\,\widetilde{\mathbf{K}}_{\mathbf{X},\mathbf{X}}\,\widetilde{\mathbf{W}}^\top\right] - \left[\widetilde{\mathbf{W}}\,\widetilde{\mathbf{K}}_{\mathbf{X},\mathbf{X}}\,\widetilde{\mathbf{W}}^\top\right]\left[\Phi + \left[\widetilde{\mathbf{W}}\,\widetilde{\mathbf{K}}_{\mathbf{X},\mathbf{X}}\,\widetilde{\mathbf{W}}^\top\right]\right]^{-1}\left[\widetilde{\mathbf{W}}\,\widetilde{\mathbf{K}}_{\mathbf{X},\mathbf{X}}\,\widetilde{\mathbf{W}}^\top\right] \\
&= \widetilde{\mathbf{W}}\left[\widetilde{\mathbf{K}}_{\mathbf{X},\mathbf{X}} - \widetilde{\mathbf{K}}_{\mathbf{X},\mathbf{X}}\,\widetilde{\mathbf{W}}^\top\left[\Phi + \widetilde{\mathbf{W}}\,\widetilde{\mathbf{K}}_{\mathbf{X},\mathbf{X}}\,\widetilde{\mathbf{W}}^\top\right]^{-1}\widetilde{\mathbf{W}}\,\widetilde{\mathbf{K}}_{\mathbf{X},\mathbf{X}}\right]\widetilde{\mathbf{W}}^\top \\
&= \widetilde{\mathbf{W}}\left[\widetilde{\mathbf{W}}\,\Phi^{-1}\,\widetilde{\mathbf{W}}^\top + \widetilde{\mathbf{K}}_{\mathbf{X},\mathbf{X}}^{-1}\right]^{-1}\widetilde{\mathbf{W}}^\top
\end{aligned}
\tag{65}
$$

and the mean can be expressed as

$$
\begin{aligned}
\mu_{\mathbf{F}\,|\,\mathbf{Y}} &= \Sigma_{\mathbf{F}\,|\,\mathbf{Y}}\,\Phi^{-1}\mathbf{Y} \\
&= \widetilde{\mathbf{W}}\left[\widetilde{\mathbf{W}}\,\Phi^{-1}\,\widetilde{\mathbf{W}}^\top + \widetilde{\mathbf{K}}_{\mathbf{X},\mathbf{X}}^{-1}\right]^{-1}\widetilde{\mathbf{W}}^\top\,\Phi^{-1}\mathbf{Y}.
\end{aligned}
\tag{66}
$$

Now we can immediately read off the posterior $p(\widetilde{\mathbf{f}}\,|\,\mathbf{Y})$ as $p(\mathbf{F}(\mathbf{X})\,|\,\mathbf{Y}) = p(\widetilde{\mathbf{W}}\,\widetilde{\mathbf{f}}\,|\,\mathbf{Y})$ is simply a transformed version

$$
\begin{aligned}
p(\widetilde{\mathbf{f}}\,|\,\mathbf{Y}) &= \mathrm{N}\left(\widetilde{\mathbf{f}}\;\middle|\;\left[\widetilde{\mathbf{W}}\,\Phi^{-1}\,\widetilde{\mathbf{W}}^\top + \widetilde{\mathbf{K}}_{\mathbf{X},\mathbf{X}}^{-1}\right]^{-1}\widetilde{\mathbf{W}}^\top\,\Phi^{-1}\mathbf{Y},\;\left[\widetilde{\mathbf{W}}\,\Phi^{-1}\,\widetilde{\mathbf{W}}^\top + \widetilde{\mathbf{K}}_{\mathbf{X},\mathbf{X}}^{-1}\right]^{-1}\right) \\
&= \mathrm{N}\left(\widetilde{\mathbf{f}}\;\middle|\;\mu_{\widetilde{\mathbf{f}}\,|\,\mathbf{Y}},\,\Sigma_{\widetilde{\mathbf{f}}\,|\,\mathbf{Y}}\right)
\end{aligned}
\tag{67}
$$

whose natural parameters are

$$
\boldsymbol{\lambda}_{\widetilde{\mathbf{f}}\,|\,\mathbf{Y}} = \left[\widetilde{\mathbf{W}}^\top\,\Phi^{-1}\mathbf{Y},\,-\frac{1}{2}\widetilde{\mathbf{W}}\,\Phi^{-1}\,\widetilde{\mathbf{W}}^\top - \frac{1}{2}\widetilde{\mathbf{K}}_{\mathbf{X},\mathbf{X}}^{-1}\right]^\top.
\tag{68}
$$

We now derive the closed form expression of the natural gradient update with a learning rate of 1, and show that it recovers $\boldsymbol{\lambda}_{\widetilde{\mathbf{f}}\,|\,\mathbf{Y}}$. The expected log likelihood (ELL) is

$$
\begin{aligned}
\mathrm{ELL} &= \mathbb{E}_{q(\widetilde{\mathbf{f}})}\left[\log \mathrm{N}\left(\mathbf{Y}\;\middle|\;\widetilde{\mathbf{W}}\,\widetilde{\mathbf{f}},\,\Phi\right)\right] \\
&= \log \mathrm{N}\left(\mathbf{Y}\;\middle|\;\widetilde{\mathbf{W}}\,\widetilde{\mathbf{f}},\,\Phi\right) - \frac{1}{2}\,\mathrm{Tr}\left[\Phi^{-1}\,\widetilde{\mathbf{W}}\,\mathbf{S}\,\widetilde{\mathbf{W}}^\top\right].
\end{aligned}
\tag{69}
$$

The required derivatives are

$$
\begin{aligned}
\frac{\partial \mathrm{ELL}}{\partial \mathbf{m}} &= -\frac{1}{2}\frac{\partial}{\partial \mathbf{m}}\left[(\mathbf{Y} - \widetilde{\mathbf{W}}\,\mathbf{m})^\top\,\Phi^{-1}\,(\mathbf{Y} - \widetilde{\mathbf{W}}\,\mathbf{m})\right] \\
&= \widetilde{\mathbf{W}}^\top\,\Phi^{-1}\,(\mathbf{Y} - \widetilde{\mathbf{W}}\,\mathbf{m})
\end{aligned}
\tag{70}
$$

where the last follows because $\Phi$ is symmetric and

$$
\begin{aligned}
\frac{\partial \mathrm{ELL}}{\partial \mathbf{S}} &= -\frac{1}{2}\frac{\partial}{\partial \mathbf{S}}\left[\mathrm{Tr}\left[\Phi^{-1}\,\widetilde{\mathbf{W}}\,\mathbf{S}\,\widetilde{\mathbf{W}}^\top\right]\right] \\
&= -\frac{1}{2}\,\widetilde{\mathbf{W}}\,\Phi^{-1}\,\widetilde{\mathbf{W}}^\top.
\end{aligned}
\tag{71}
$$

The natural gradient is now given as

$$
\begin{aligned}
\frac{\partial \mathrm{ELL}}{\partial \boldsymbol{\mu}_{\widetilde{\mathbf{f}}\,|\,\mathbf{Y}}} &= \begin{bmatrix}\frac{\partial \mathrm{ELL}}{\partial \mathbf{m}} - 2\frac{\partial \mathrm{ELL}}{\partial \mathbf{S}}^\top\mathbf{m} \\ \frac{\partial \mathrm{ELL}}{\partial \mathbf{S}}\end{bmatrix} = \begin{bmatrix}\widetilde{\mathbf{W}}^\top\,\Phi^{-1}\,(\mathbf{Y} - \widetilde{\mathbf{W}}\,\mathbf{m}) - \widetilde{\mathbf{W}}^\top\,\Phi^{-1}\,\widetilde{\mathbf{W}}\,\mathbf{m} \\ -\frac{1}{2}\,\widetilde{\mathbf{W}}\,\Phi^{-1}\,\widetilde{\mathbf{W}}^\top\end{bmatrix} \\
&= \begin{bmatrix}\widetilde{\mathbf{W}}^\top\,\Phi^{-1}\,\mathbf{Y} \\ -\frac{1}{2}\,\widetilde{\mathbf{W}}\,\Phi^{-1}\,\widetilde{\mathbf{W}}^\top\end{bmatrix}.
\end{aligned}
\tag{72}
$$

The natural gradient update with a learning rate of 1 is

$$
\boldsymbol{\lambda}_{q(\widetilde{\mathbf{f}})} = \frac{\partial \mathrm{ELL}}{\partial \boldsymbol{\mu}_{\widetilde{\mathbf{f}}\,|\,\mathbf{Y}}} + \boldsymbol{\lambda}_{p(\widetilde{\mathbf{f}})}
\tag{73}
$$

where $\boldsymbol{\lambda}_{p(\widetilde{\mathbf{f}})} = \left[\mathbf{0},\,-\frac{1}{2}\mathbf{K}^{-1}\right]^\top$ are the natural parameters of the prior $p(\widetilde{\mathbf{f}})$, hence after the update the natural parameters are

$$
\boldsymbol{\lambda}_{q(\widetilde{\mathbf{f}})} = \begin{bmatrix}\widetilde{\mathbf{W}}^\top\,\Phi^{-1}\,\mathbf{Y} \\ -\frac{1}{2}\,\widetilde{\mathbf{W}}\,\Phi^{-1}\,\widetilde{\mathbf{W}}^\top - \frac{1}{2}\mathbf{K}^{-1}\end{bmatrix}.
\tag{74}
$$

which recover those of $p(\widetilde{\mathbf{f}}\,|\,\mathbf{Y})$, and hence we recover the optimal posterior. $\qquad\square$

# B  Further Experimental Details and Results

EKS methods were run on CPUs. State-space methods running on GPU used the parallel form of the Kalman smoother (see [55, 20]).

## B.1  An extension of AUTOIP

If one drops the requirement for state-space representations then the approximations proposed in Sec. 4 directly define approximations to the variational GP defined by Eqn. (13), and hence directly extend AUTOIP. For example on the non-linear damped pendulum in Sec. 7 we run this extension of AUTOIP with whitening and 50 inducing points for $C = 1000$ and achieve an RMSE of $0.06 \pm 0.001$ and running time of $158.16 \pm 0.34$, clearly improving the running time against AUTOIP. However the benefit of our methods is that PHYSS-GP remains linear in temporal dimensions which is vital for applications that are highly structured in time [20].

## B.2  Modelling Unknown Physics

Modelling of missing physics can be handled by parameterising unknown terms with GPs. For example take a simple non-linear pendulum

$$\frac{d^2\theta}{dt^2} + \sin(\theta) = 0. \tag{75}$$

Now consider that the the $\sin(\theta)$ is unknown and we would like to learn it. If we define the our differential equation in Eqn. (4) as

$$g = \frac{d^2 f_1}{dt^2} + f_2(t) = 0 \tag{76}$$

where both $f_1(\cdot), f_2(\cdot)$ are latent GPs that we wish to learn. We now construct 300 observations for training from the solution of Eqn. (75) across the range $[0, 30]$ and 1000 for testing. We run PHYSS-GP and compare the similarity of the learnt latent GP $f_2(\cdot)$ to the true function at the test locations and achieve an RMSE of 0.068 indicating we have recovered the latent force/unknown physics well.

## B.3  Monotonic Timeseries

This first example showcases the effectiveness of PHYSS-GP in learning monotonic functions. Monotonicity information is expressed by regularising the first derivative to be positive at a set of collocation points [51]:

$$p(\mathbf{Y} \mid \mathbf{f}) = \mathrm{N}\left(\mathbf{Y} \mid \mathbf{f}, \sigma_y^2\right), \; p(\mathbf{0} \mid \frac{\partial \mathbf{f}}{\partial t}) \; = \Phi(\frac{\partial \mathbf{f}}{\partial t} \cdot \frac{1}{v})$$

where $\Phi(\cdot)$ is a Gaussian cumulative distribution function, and $v = 1e - 1$ is a tuning parameter that controls the steepness of the step function. We plot predictive distributions of (batch) GP and PHYSS-GP in Fig. 5. The GP fits data and does not learn a monotonic function. However, using 300 collocation points, PHYSS-GP is able to include the additional information and learn a monotonic function whilst running 1.5 times faster.

## B.4  Non-linear Damped Pendulum

All models were run using an Nvidia Titan RTX GPU and an Intel Core i5 CPU. All were optimised for 1000 epochs using Adam [32] with a learning rate of 0.01. Both the GP and AUTOIP had an RBF kernel (following Long et al. [40]) and PHYSS-GP used a Matérn-7⁄2; all with a lengthscale of 1.0. The observation noise was initialised to 0.01 and the collocation 0.001. Both were fixed for the first $40\%$ of training and then released. Predictive distribution of PHYSS-GP and AUTOIP are plotted in Fig. 6.

## B.5  Curl-free Magnetic Field Strength

All models were run using an Nvidia Titan RTX GPU and an Intel Core i5 CPU. All models are run for 5000 epochs using Adam with a learning rate of 0.01, and use a Matérn-3⁄2 kernel on time,

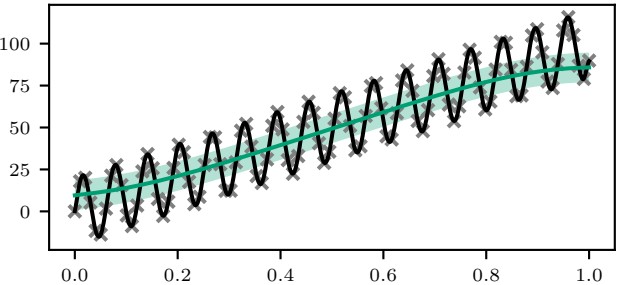

Figure 5: Predictive distributions of GP and PHYSS-GP on the monotonic function in App. B. The GP cannot incorporate monotonicity information and fits the data.

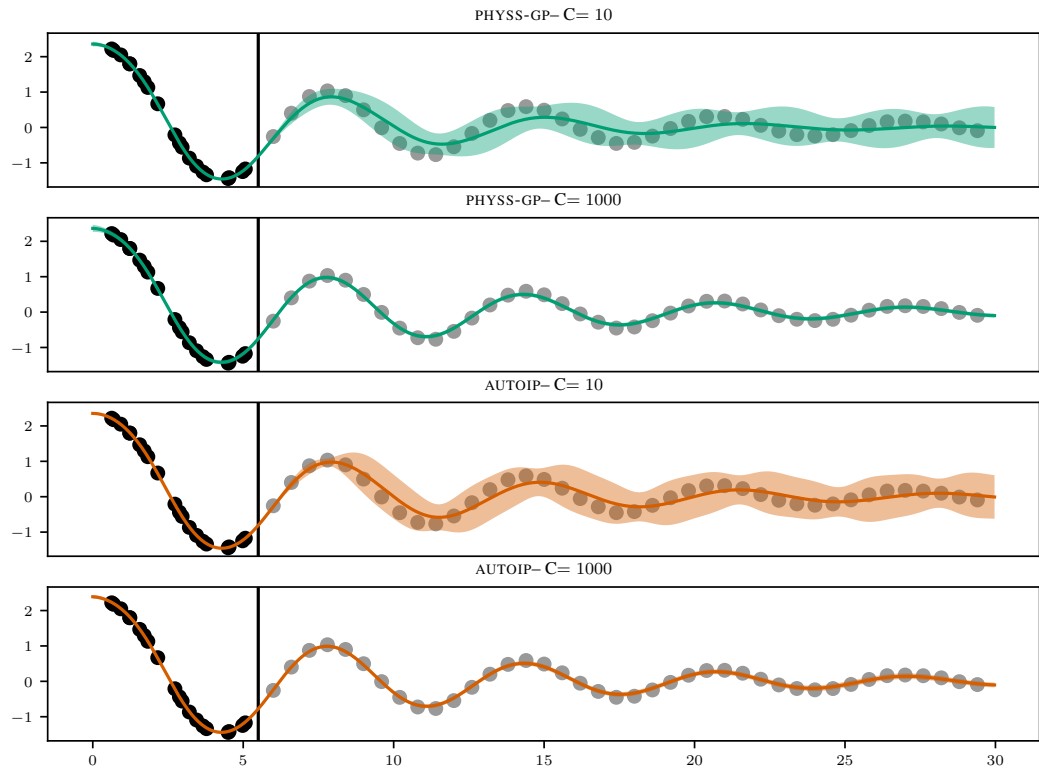

Figure 6: Predictive distributions on the Damped Pendulum.

with ARD RBF kernels on the spatial dimensions, with a lengthscale of $0.1$ across all. The Gaussian likelihood is initialised with a variance of $0.01$ and held for $40\%$ of training. All our methods used a natural gradient learning rate of $1.0$ as this is the conjugate setting.

## B.6 Diffusion-Reaction System

We use data provided by [49] under an MIT license. All models were run using an Nvidia Titan RTX GPU and an Intel Core i5 CPU. Our method PHYSS-SVGP and PHYSS-SVGP$_H$ use a Matérn 72 kernel on time and an RBF of space, both initialised with a lengthscale of $0.1$. We place the collocation points on a regular grid of size $20 \times 10$ and use $M_s = 20$ spatial inducing points. We pretrain for 100 iterations using a natural gradient learning rate of $0.01$ and after use a learning rate $0.1$ for the remaining 19000 iterations. AUTOIP uses a RBF kernel on both time and space with a lengthscale of $0.1$. We place the collocation points on a regular grid of size $10 \times 10$. All models use Adam with a learning rate $0.001$ and train for a total of 20000 iterations.

### B.7 Ocean Currents

Our method PHYSS-SVGP$_H$ was run using an Nvidia Titan RTX GPU and an Intel Core i5 CPU. We ran for 10000 iterations, using Adam with a learning rate of 0.01. For natural gradients with used a learning rate of 0.1. We used a Matérn-3⁄2 kernel on time and RBF kernels on both spatial dimensions with lengthscales $[24.0, 1.0, 1.0]$. We used 100 spatial inducing points and a spatial mini-batch size of 10.

