# OpenReview forum: "Physics-Informed Variational State-Space Gaussian Processes"
_NeurIPS.cc/2024/Conference — NeurIPS 2024 poster_

### Official Review · Reviewer_BaEt · 2024-07-05

**Soundness:** 3
**Presentation:** 3
**Contribution:** 2
**Rating:** 6
**Confidence:** 2

**Summary:**

The paper introduces a method for training spatio-temporal Gaussian processes which incorporate physics constraints including satisfying the governing equations at a number of collocation points and satisfying curl / divergence free constraints. Their approach scales both linearly in time and space by leveraging a number of approximations.

**Strengths:**

- While many of the individual components (such as incorporating curl / divergence free constraints and satisfying governing equations at a number of collocation points) have been done before, I believe the combination of ideas here is novel.
- Showing how to achieve linear complexity in the space and time dimension is valuable.
- Showing how your approach recovers some prior works as a special case from a more general perspective is also valuable.
- The bevy of test cases demonstrating the efficacy of your approach on both synthetic and real-world problems convincingly demonstrate the advantages offered by your approach.

**Weaknesses:**

- The problem statement is poorly motivated in my perspective. It would be helpful to be more specific about what you hope to achieve with hybrid modeling. For example, is your goal to improve the predictive accuracy of mechanistic models by incorporating data? What computational efficiency do you hope to improve? i.e. do you want to reduce the amount of data needed to train physics surrogates? Do you argue your approach is more efficient than solving the mechanistic model using traditional methods? etc. While I understand space is limited, being more specific about the problem statement will also help guide more targeted numerical studies in future works.
- I think you need slightly more of a discussion on virtual observations (see questions below). When virtual observations are competing with actual data it would be helpful to discuss in detail how you think about $\epsilon_C$ and choosing the number of collocation points.
- Since you state that one of the goals of your approach is to be useful in uncertainty quantification, the numerical studies would have been strengthened by comparing more than just RMSE. For example, comparing the continuous ranked probability score (CRPS) would have given an indication of how well your approach is estimating uncertainty.
- Minor:
  - L94 space cases -> special cases?

**Questions:**

- As I understand it, you can always choose to satisfy the differential equations at enough collocation points such that you overwhelm the data. For example, say I have very limited data so I choose the number of collocation points to be much greater than the number of data points. Clearly, if I use too many collocation points, my model will ignore the data and just follow the differential equations. How do you think about this perspective in the context of uncertainty quantification?
- As a follow up to this previous point, what are the implications of assuming that $0_n^{(C)} = g(F_n) + \epsilon_C$? Is it correct to think about this as implicitly assuming that the PDE is stochastic?
- Are there situations where you believe the Gaussian process assumption could be limiting?

**Limitations:**

I think the authors have done a good job of identifying some potential limitations. I think the paper would greatly benefit from a more in depth discussion on using collocation points to enforce governing equation constraints and the limitations this perspective brings.

---

> ### Author Rebuttal · Authors · 2024-08-06
>
> Thank you for your review. We address the points you raised below.
>
> **W1: Problem statement.**
> Our work contributes to the growing field of data-driven physics-informed models. These are hybrid models that aim to exploit physical inductive biases and data observations. Recently there have been principled ways to approach this problem through GPs, where the current state of the art includes AUTOIP and HELMHOLTZ-GP.  However, these approaches are cubic in the number of data points limiting them to small-scale studies. In this work we provide a unifying view of such GP approaches and derive state-space algorithms that are linear in the temporal dimension, bringing computational complexity closer to standard ODE solvers, and enable the application to large scale spatio-temporal problems.
>
> **W2: Virtual observations.**
> Please, see Q1 below.
>
> **W3: Uncertainty quantification.**
> We agree that providing uncertainty metrics is valuable. In Table 1, we have already provided negative log predictive distributions (NLPDs) however we will report them and CRPS for all experiments. For the Diffusion-Reaction experiment they are
>
> | Model    | RMSE | NLPD | CRPS |
> | -------- | ------- | ------- | ------- |
> | PHYSS-EKS  |  $0.06$  | $-1.26$ | 0.038 |
> | PHYSS-SVGP$_{H}$ | $0.17$ | $1.69$ | 0.077 |
> | AUTOIP    | $0.17$    | $-0.29$ | 0.065 |
>
>  and for the Ocean Current experiment we achieve an NLPD of $-0.52$ and CRPS of $0.078$.
>
> **W4: Typo.**
> Thank you for pointing out this typo, we have addressed this now in the manuscript.
>
> **Q1: Uncertainty.**
> In this paper, we assume that the data observations and mechanistic descriptions align (although we can simply extend to handle missing physics, see review zAxE Q2). In this setting, we can view the uncertainty as representing our beliefs on 'how well we have solved the system'. Since we can only ever place a finite number of colocation points there will always be some error in our solution, and uncertainty quantification is vital in representing that. Moving forward having capable (both computationally and predictively) uncertainty-equipped models will be vital for handling misspecification (perhaps in prior or the likelihood) and for parameter learning.
>
> **Q2: Collocation points?**
> Please, see the discussion in reply to reviewer **zAxE** Q3.
>
> **Q3: GP as a limitation.**
> Under non-linear differential equations the true resulting posterior (of the generative model in Eqs. 5-6) will be non-Gaussian, and with highly non-linear equations (and or chaotic systems) approximating the true posterior with a Gaussian could lead to underestimation of the uncertainty (Turner et al., 2011). However, for many problems, the Gaussian assumption is effective as a Gaussian system is much simpler to solve [54,22].
>
> - Richard Turner and Maneesh Sahani (2011). Two problems with variational expectation maximisation for time-series models. *Bayesian Time Series Models.*

---

> > ### Comment · Reviewer_BaEt · 2024-08-08
> >
> > Thank you for your thoughtful response.
> >
> > Regarding the W1, let me try to rephrase. I agree that your unifying view of GP
> > approaches for hybrid modeling is itself a valuable contribution. In my understanding
> > of physics informed learning, it is often useful to understand the downstream tasks
> > which you hope to achieve using a specific approach as this will guide algorithm design.
> > For example, some approaches may seek to build predictive models
> > which can learn from less data because they
> > explicitly incorporate physics knowledge. Other approaches may seek to improve the
> > accuracy of flawed mechanistic models using data. Other approaches may seek to
> > use data to approximately solve differential equations more quickly
> > with error estimates. Other approaches may seek to build cheap, approximate models
> > of differential equations so that they can be used in optimization and control.
> > My point with W1 was that I think it would be very helpful to discuss where you
> > think your approach fits into the broader context of physics informed ML.
> >
> > Thank you for providing CRPS scores. I think this will be helpful for those
> > hoping to extend/compare to your work in the future.
> >
> > Regarding Q1 (and this relates back to W1), thank you for clarifying. I think making it
> > clear in the main text that the Gaussian likelihood is just a relaxation of a "delta"
> > likelihood is helpful in understanding the motivations of your approach.

---

> > > ### Author Response · Authors · 2024-08-11
> > >
> > > Thank you for your comment and for clarifying W1.
> > >
> > > Our proposed approach fits into building probabilistic models where mechanistic/physics knowledge is incorporated as an inductive bias to improve performance (such as the latent quantity of interest should follow a differential equation, or the vector field we are modelling should be curl-free). We agree that this allows one to use less data because the physics has been explicitly incorporated. However our approach can also be extended to the setting where there is partial physical knowledge, see discussion Q2 in reply to zAxE.  We do not see this work as an alternative to classical PDE solvers. We will add this discussion to our introduction and agree that it will help better place our work within the wider field of physics-informed ML.

---

### Official Review · Reviewer_zAxE · 2024-07-09

**Soundness:** 3
**Presentation:** 3
**Contribution:** 3
**Rating:** 6
**Confidence:** 3

**Summary:**

The paper introduces a formulation for state-space GP which aims to capture the behaviour of PDE-based systems. The paper suggests multiple techniques to speed up the GP inference, including through decomposed kernel, variational methods on natural parameterisation of distribution, use of inducing points

**Strengths:**

The paper provides good motivation and use cases for physics-informed GPs, and the need for scaling the inference process. The paper also provides adequate results and demonstrations of the GP for prediction and uncertainty quantifications, which well motivates the benefits of GPs over other solver types.

**Weaknesses:**

- The paper may be a bit hard to follow if one is not familiar with GPs already. For example, it may be difficult to immediately see how the PDE or the BV are incorporated into the probability distribution during the inference stage.
- It may also be arguable that some parts of the paper can be seen as just application of standard sparse GP techniques and variational methods into the context of physics-informed GPs.

**Questions:**

- Are there any intuition as to why the decomposition of the kernel is done between the spatial and the temporal variables? Is it for computational reasons or can it be linked to properties of physical phenomena?
- Does (or can) the method perform in the case that the full form of the PDE is unknown? Or is the paper mostly focused on cases of just generating the PDE solution?
- In the formulation given by Eq 6, the PDE and BC residuals are assumed to arise from a Gaussian distribution. How can they be related to the physical realisation of the PDE, or how can they be interpreted properly as a probability distribution? How realistic is this assumption?
- Are there any additional assumptions in the form of g(f) should take for this method to work?

**Limitations:**

Limitations are adequately mentioned.

---

> ### Author Rebuttal · Authors · 2024-08-06
>
> Thank you for your review and positive view of our work. We address the points you raised below.
>
> **W1: Clarity.**
> To aid understanding we will provide an additional notation section and nomenclature table in the appendix as well as the proposed additions mentioned in the reply to reviewer 4ngK. To clarify, the boundary values are incorporated as observations (as defined in Eq. 6). At the inference stage this is incorporated through an additional likelihood. This means for  PHYSS-VGP the likelihood term in Eq. (13) can be written as $p(Y_n | \bar{f}) = p(Y^\mathcal{O} | H_\mathcal{O} \, F_n) \, p(O | g(F_n)) \, p(Y^\mathcal{B} | H_\mathcal{B} \, F_n)$ where the differential equation is used to define $g$. We will add this discussion into Sec. 3.
>
>
>
>
> **W2: Novelty.**
> Please, see the response to reviewer X3gM W1.
>
> **Q1: Decomposition of kernels.**
> The primary motivation for the decomposition of the kernel into space and time is computation. This is a standard assumption for GPs and in this paper, we show that we can exploit this separability to derive state-space inference algorithms that are linear in the temporal dimension. One would expect that for linear and autonomous differential equations (time-invariant in this case) this will be a reasonable assumption, however, this is an interesting avenue for future research.
>
> **Q2: Unknown physics.**
> Thank you for this excellent question. This paper has focused on modelling solutions to differential equations; however, we can simply extend our framework (as in [35]) to model missing physics through GPs. This is similar to the latent force model of [66] except we now jointly model the latent force and the solution as GPs. To demonstrate this we construct 300 observations from a non-linear pendulum $\frac{d^2 \theta}{dt^2} + \sin(\theta) = 0$. We will consider the $\sin(\theta)$ term as missing, which we would like to learn. We now define $g=\frac{d^2 \, f_1}{dt^2} + f_2(t) = 0$. We achieve an RMSE of $0.068$ indicating we have recovered the latent force/unknown physics well. Going beyond this simple example is an exciting avenue for future work. We will add this discussion to the main paper with this example in the appendix.
>
> **Q3: PDE residual.**
> This is a common modelling assumption for collocation based methods. Ideally, we would model the residual as a noise-free observation (arising from a delta likelihood) and enforce it exactly. Treating the residual as arising from a Gaussian likelihood is a relaxation of this. We will add this discussion to the main paper.
>
> **Q4: Assumptions on $g$.**
> In general, for the probabilistic model to be well defined we require that g is measurable. To ensure that we are modelling something 'useful' we require that there be a unique well-defined solution to the differential equation. For ODEs, one can appeal to the Picard-Lindelöf theorem (see Thm. 36.4 in [22]) which gives general conditions for unique solutions to exist and requires that the non-linear operator be locally Lipsitz continuous. However, we are unaware of any results for general PDEs. Extending to problems that have multiple solutions would be an interesting avenue for future work. We will add this discussion to the main paper.

---

### Official Review · Reviewer_4ngK · 2024-07-11

**Soundness:** 1
**Presentation:** 1
**Contribution:** 2
**Rating:** 4
**Confidence:** 2

**Summary:**

The paper introduces a novel approach for solving partial differential equations (PDEs) using a Gaussian process prior. It leverages established methods for approximate inference and provides a cohesive framework that unifies existing results in probabilistic differential equation solving.

**Strengths:**

The paper introduces a novel approach for solving partial differential equations (PDEs) using a Gaussian process prior. It leverages established methods for approximate inference and provides a cohesive framework that unifies existing results in probabilistic differential equation solving.

**Weaknesses:**

The paper’s presentation is challenging to follow. Despite my familiarity with continuous-time filtering, smoothing, Gaussian process (GP) literature, and continuous-time stochastic optimal control (where some of these partial differential equations (PDEs) appear), I struggle to determine the paper’s solidity. Numerous mathematical inaccuracies and inconsistencies hinder comprehension. For instance, the abrupt switch between notation for space variables (e.g., $s$ and $t$) and the combined space-time variable ($x$) lacks clear definition. While I recognize that machine learning sometimes necessitates complex notation, in this case, it impedes understanding.

The main model section exacerbates the issue. Equations 5 and 6 remain elusive; the dimensions of $Q$ and $P$ are unspecified. The definition of $F_n$ confuses me, as it prevents a sensible matrix product computation. Additionally, the collocation point definition remains unclear. Even in Example 3.1, the meaning of $g_k$ remains unknown to me. Consequently, assessing the paper’s solidity proves challenging.
In my view, this paper requires substantial revision before acceptance. Although I appreciate the effort invested, from a reviewer’s standpoint, the current presentation poses significant hurdles.

**Questions:**

- In the background section please define N and F. And the relations between al the other variables N_t, N_s x_{t,s}, y_{t,s}, d, etc. I find this very confusing – is there also a clash in notation?
-	In Eq (3) can you elaborate if $\mathbf f$ is multidimensional and how does it relate to $f$. This notation is again used in line 78.
-	In line 66 is this definition correct? Somehow I have a hard time coming up with the correct dimensions.
-	In line 70 $\bar{\mathbf f} $ is a time depdent function. But in line 78 it is a space time depdent function. Could you elaborate on this?
-	In Line 87 is the non-linear differential operator only dependent on space derivatives, or can the also be higher order time derivatives?
-	Eq (5): $\bar f_q(\mathbf X_n)$ should be a vector of dimensionality D, but in line 98 W is of dimensionality $PD \times QD$. Can you elaborate on this.
-	In eq(7) can you elaborate what $g(\mathbf F_n)$ is? Is it an element-wise operation for each element in $\mathbf F_n$?
-	In line 265-266 can you please elaborate what $z$ is?

**Limitations:**

-

---

> ### Author Rebuttal · Authors · 2024-08-06
>
> We thank you for your review and for raising questions that have helped improve the presentation of the paper.
>
> We understand your concern related to the complicated notation. To aid understanding we will use the additional page in the camera-ready stage to include a notation section that will clarify the dimensions of all quantities. Additionally, we will provide a table in the appendix reporting the same. However, we would like to emphasize that throughout the paper we have attempted to use standard notation.
>
> **W1: Definition of Q/P.**
> Q/P are scalars denoting the number of latent functions and outputs. Q as the number of latent functions is established notation in multi-output GPs (see, e.g., [67, 37]). Additionally in the App. A.3 we have already provided an expanded form which provides the dimensions of all the relevant quantities. We will explicitly state these dimensions in Sec. 3 of the main paper.
>
> **W2: Abrupt switch in s/t.**
> We are unsure what you mean by this. Throughout we have been clear that we are operating in the spatio-temporal setting (explicitly stated on line 58/59) and throughout 't' refers to time and 's' to space.
>
> **W3: Collocation point definition.**
> The collocation points are defined in Eq. (6). Intuitively we would want the function that we learn (which we have placed a GP prior over) to coincide exactly with the solution of the differential equation at hand, i.e. that the residual between them is zero. In practice, we can only ever enforce this at a finite set of locations, and these are called the collocation points.
>
> **W4: Definition of $g$.**
> The function $g$ is computing the residual described in [Collocation point definition] above. It measures the (pointwise) error between the current function and the solution to the differential equation. Equivalently if $f$ followed the differential equation exactly, then, by definition, its first time derivate would equal $N_\theta(f)$ and $g(f)$ would be zero.
>
>
> **Q1: Definitions.**
> All quantities $N_s, N_t, x_{t,s}, y_{t, s}$ are defined on line 59. Again this is in accordance with the state-space GP literature (see [18]).
>
> **Q2: Multi-output.**
> As explained on line 79, Eq. (3) is a multi-output prior over a latent function f and its space/time derivates. Here we use established notation (see [47]) where a sample from $\bar{f}$ is of dimension $N \cdot D$ and corresponds to the multiple-outputs being stacked. We will clarify this further in the main paper.
>
> **Q3: Dimensions.**
> Yes, the equation is correct. As described on line 65/66 this is a $d$-dimensional vector (where $d$ is the number of time-derivates) and $\bar{f}$ is the corresponding state vector. See for example [48].
>
> **Q4: Notation.**
> Throughout we use $\bar{f}$ to denote $f$ together with its spatial and/or temporal derivates. On lines 66-71 we highlight that standard state-space GPs construct a state over $f$ and its time derivates, and hence use the notation $\bar{f}$. Specifically on line 66 we discuss timeseries models, and hence the state only depends on time. On lines 70/71 we discuss how spatio-temporal GPs can be represented as a state-space model, where now the state is modelling the temporal dynamics at each spatial point and hence the state is constructed over the spatial points (see [48, Sec. 12.5]). On line 78 we are constructing a GP prior over $\bar{f}$ at all the input locations $X$, as defined in Eq. (3). We will further clarify this in Sec. 2 of the main paper.
>
> **Q5: Higher-order derivatives.**
> The non-linear operator can be dependent on any order of spatial or temporal derivatives. The only requirement is that the latent multi-output GP is defined over those derivatives as well (which will require sufficiently smooth temporal and spatial kernels). We will add this discussion to Sec. 3 to clarify.
>
> **Q6: Dimensionality.**
> As discussed in (1-1) above this follows standard multi-output GP notation. Here $f_q$ are (stacked) multi-ouput GPs. At a single input location a sample will have dimension $D$ (as defined on line 73). $F_n$ is defined by stacking all outputs from the $f_q$ GPs, which for a single input location will have dimension $Q \times D$. This is then linearly mixed to create $P \times D$ outputs. To make this clearer we will write the stacking explicitly so that $F_n (X_n) = W \, \text{vec}[f_1(X_n), \cdots, f_Q(X_n)]^\top$ and state all these dimensions in Sec. 3 of the main paper.
>
> **Q7: What is $g$ doing?**
> The function $g: \mathbb{R}^{PD} \to \mathbb{R}$ computes the residual between the latent process and the differential equation (see W4. in **4ngK**). This follows the notation set out in [22, 42, 35].
>
> **Q8: What is z?**
> The full magnetic field is defined in a 3-dimensional space on which we place our spatio-temporal model PHYSS-GP. Here 'z' is simply a label to denote the third dimension and can be thought of as 'depth'. We will be more explicit about this.

---

> > ### Comment · Reviewer_4ngK · 2024-08-11
> >
> > I have read the rebutal and thank the authors for their answers. I do not have any further questions. In view of the other reviews and the promised improvement in clarification of the paper, I will increase my score by one point, though I am still of the opinion that the is work hard to follow.

---

> > > ### Author Response · Authors · 2024-08-11
> > >
> > > We thank you for your comment and your score increase.

---

### Official Review · Reviewer_z1FQ · 2024-07-12

**Soundness:** 3
**Presentation:** 3
**Contribution:** 3
**Rating:** 7
**Confidence:** 3

**Summary:**

In this paper, the authors present a physics-informed Gaussian process based approach to learn the solution of ODE and PDE systems. In particular, they address the challenge of the cubic computational complexity with respect to the number of spatial observations. It is shown that multiple state-of-the-art approaches can be recovered as special cases of the presented approach, i.e., it is a unifying framework. With additional approximation techniques, the cubic spatial computational cost is reduced to linear. In multiple simulations, the proposed method achieves similar or better results compared to SOTA methods but with a significant reduced computational time.

**Strengths:**

- The proposed method demonstrates a significant improvement of the computational complexity with respect to the number of spatial observations compared to SOTA methods.
- The beauty and originality of the proposed model is its unifying property, including existing approaches such as HelmholzGP and AUTOIP.
- The techniques and ideas in the paper are clearly presented. I really enjoyed reading the paper.

**Weaknesses:**

- There are some typos such as line 103 “develop sate space algorithms for”, line 167 “complexity of O(N  (N_s” (missing t), and eq (17) Nt instead of N_t. Furthermore, I think that line 257  “RMSE of all models increases as the number of collocation points increases” make no sense as the RMSE is decreasing. Please check the paper carefully.
- In the experimental section, some phenomena are rarely discussed. For instance, the RMSE of PHYSS-GP is increasing with 1000 collocation points.

**Questions:**

- I assume that there exist PINN based approaches with some form of uncertainty quantification. It would be interesting to compare the GP-based method to NN-based approaches (not necessarily for this paper but in general). Do you have any thoughts on that?
- Is it possible to use the same approximations in AUTOIP to reduce the cubic computational cost? If so, is there still a benefit of the proposed method?

**Limitations:**

Yes

---

> ### Author Rebuttal · Authors · 2024-08-06
>
> Thank you for your review and positive view of our work. We address the points you raised below.
>
> **W1: Typos.**
> Thank you for highlighting these typos, we have addressed them all. However, line 167 is *not a typo*; the computational complexity is $O(N (N_s \, d_s \, d)^3)$ because the expected log-likelihood decomposes across all spatio-temporal datapoints (as shown by the first term on Eqn 17), and hence is linear in $N$. This motivates the spatial-minibatching approximation where this is reduced to be $O(N_t \, (N_s \, d_s \,d)^3)$. We will add this explanation to line 167.
>
> **W2: Increasing RMSE.**
> This is an artefact of the stochastic nature of the optimisation algorithms used for inference. We have rerun our Non-linear Damped Pendulum experiment across 5 datasets constructed with different random seeds. We report the results below.
>
> | Model    | C | Time |RMSE | NLPD |
> | -------- | ------- | ------- | ------- | ------- |
> | PHYSS-GP  | $10$ | 	$59.27 \pm 38.524$ |	$0.20 \pm 0.007$ |	$-0.28 \pm 0.102$ |
> | | $100$ |	$112.82 \pm 0.400$ |	$0.05 \pm 0.001$ |	$-0.44 \pm 0.134$ |
> | | $500$ |	$139.86 \pm 0.541$ |	$0.05 \pm 0.003$ |	$-0.79 \pm 0.414$ |
> | | $1000$ |	$122.94 \pm 40.184$ |	$0.05 \pm 0.003$ |	$-0.87 \pm 0.327$ |
> |AUTOIP  | $10$ |	$117.16 \pm 27.378$ |	$0.16 \pm 0.001$ |	$-0.41 \pm 0.070$ |
> ||	$100$ 	|$154.10 \pm 31.590$ |	$0.05 \pm 0.001$ |	$-0.51 \pm 0.129$ |
> || 	$500$ |	$1058.08 \pm 17.497$ |	$0.05 \pm 0.001$ |	$-0.88 \pm 0.087$ |
> || 	$1000$ |	$5600.90 \pm 36.231$ |	$0.05 \pm 0.001$  |	$-1.39 \pm 0.091$ |
>
>
> **Q1: Uncertainty in PINNs.**
> PINNs have become a popular method for solving differential equations and amount to a highly complicated optimisation problem that requires specialised training regimes (see Wang et al., 2024). Current approaches to quantifying uncertainty (UQ) in PINNs are based on dropout (see Papamarkou et al.) and conformal predictions (see Podina et al., 2024). In recent years UQ for deep learning has received much attention however is limited by its computational cost (see Papamarkou et al., 2024). Exciting avenues of work could be to explore combinations of state-space algorithms and PINNs, to achieve linear time complexities but with the flexibility of PINNs. We will add this discussion to the related work section of the main paper.
>
> - Podina et al. (2024). Conformalized physics-informed neural networks. In *ICLR 2024 Workshop on AI4Differential Equations In Science*.
> - Papamarkou et al. (2018). Position: Bayesian deep learning is needed in the age of large-scale AI. arXiv preprint.
> - Zhang et al. (2018). Quantifying total uncertainty in physics-informed neural networks for solving forward and inverse stochastic problems. *Journal of Computational Physics*.
> - Wang et al. (2024). Respecting causality for training physics-informed neural networks. *Computer Methods in Applied Mechanics and Engineering*.
>
> **Q2: Application to AUTOIP.**
> The derivation of the state-space algorithms hinges on the approximate posterior being represented as a posterior with likelihoods/sites that decompose across time, which is guaranteed within the natural gradient framework (as shown in Eq. 16). AUTOIP has no such guarantees since it optimises the natural parameters in Euclidean space with optimizers like Adam, and uses a whitened representation. However, if one drops the requirements of deriving a state-space algorithm, all of the approaches for reducing the spatial computational complexity in Sec. 4 can equally be applied to AUTOIP. Indeed, this is simple to do within our codebase (that will be released on publication). For example, on the Non-linear damped pendulum, we run this extension of AUTOIP with whitening and 50 inducing points for $C=1000$ and achieve an RMSE of $0.06 \pm 0.001$ and running time of $158.16 \pm 0.34$, clearly improving the running time against AUTOIP. This is still slower than our methods as it is cubic in the number of inducing derivatives/points. We will add this example to the appendix and this discussion into the main paper. The benefit of our proposed approach is that we remain linear in the temporal dimension which is vital for applications that are highly structured in time.

---

> > ### Comment · Reviewer_z1FQ · 2024-08-12
> >
> > Thank you for the clarification. My recommendation is to accept the paper.

---

### Official Review · Reviewer_X3gM · 2024-07-12

**Soundness:** 3
**Presentation:** 2
**Contribution:** 2
**Rating:** 6
**Confidence:** 4

**Summary:**

The submission discusses conditioning spatiotemporal Gaussian processes using differential equation constraints and observational data.
Specifically, the temporal component is handled by a Markovian prior (to achieve linear complexity), and the spatial component is dealt with by variational methods.
As such, the proposed algorithm extends the work by Hamelijnck et al. [18] to differential-equation constraints (and to handle spatial mini-batching); or, from the opposite perspective, it implements a version of a probabilistic numerical solver for spatiotemporal PDEs with variational methods (in space).
The resulting algorithm is evaluated on a damped pendulum ODE, a curl-free magnetic strength field, a one-dimensional reaction-diffusion PDE, and an ocean-current problem.

**Strengths:**

All components are technically sound, and the submission fits into recent literature on dynamical systems, Gaussian processes, and physics-informed machine learning.
Overall, I think this is a nice paper, and despite a few minor weaknesses (see "Weaknesses"), I recommend acceptance.

**Weaknesses:**

I identify two weaknesses:

1. The proposed algorithm's scientific novelty as a combination of known techniques is limited. Hamelijnck et al. essentially cover spatiotemporal variational inference with a Markovian prior in the temporal dimension. Conditioning Gaussian processes on differential equation constraints via collocation has become standard practice in recent years. Berlingheri et al. developed the curl- and divergence-free spatial priors. The combination of these techniques is novel, but the increment to the existing literature is relatively small.
2. The clarity of the submission's relation to existing approaches would improve by discussing more articles related to probabilistic numerical methods. Currently, the manuscript cites the PMM by Cockayne et al., the ODE (initial value problem) solvers by Schober et al., Krämer et al., and Tronarp et al., and the GP-PDE paper by Pförtner et al. (all references are in the submission). Beyond those papers, Krämer et al. (below) describe solving time-dependent, nonlinear PDEs with collocation and Markovian priors; Schmidt et al. (below) combine nonlinear collocation constraints with observational data (for ODEs); and Krämer and Hennig (below) solve boundary value problems with collocation. There are other related papers, but I believe the former articles should be discussed in the manuscript. Further, the literature on statistical finite element methods is related and should be acknowledged: see Duffin et al. (below).


The first weakness is more significant than the second one. I expect that the second weakness will be relatively straightforward to resolve.
In any case, I believe the strengths outweigh the weaknesses and lean towards recommending acceptance.


**References:**

Schmidt, Jonathan, Nicholas Krämer, and Philipp Hennig. "A probabilistic state space model for joint inference from differential equations and data." Advances in Neural Information Processing Systems 34 (2021): 12374-12385.

Krämer, Nicholas, Jonathan Schmidt, and Philipp Hennig. "Probabilistic numerical method of lines for time-dependent partial differential equations." International Conference on Artificial Intelligence and Statistics. PMLR, 2022.

Krämer, Nicholas, and Philipp Hennig. "Linear-time probabilistic solution of boundary value problems." Advances in Neural Information Processing Systems 34 (2021): 11160-11171.

Duffin, Connor, et al. "Statistical finite elements for misspecified models." Proceedings of the National Academy of Sciences 118.2 (2021): e2015006118.

**Questions:**

None

**Limitations:**

Discussed.

---

> ### Author Rebuttal · Authors · 2024-08-06
>
> Thank you for your positive review. We address your two comments below.
>
> **W1: Novelty of the proposed approach.**
> Whilst we agree that some special cases of our framework are known, the 'beauty and originality of the proposed framework is its unifying property' (reviewer **z1FQ**). Not only do we provide a variational state-space framework unifying AUTOIP, HELMHOLTZ-GP, PMM, and EKS (Thm. 1), but we additionally extend this framework in three ways
>     - Spatial-temporal derivative inducing points which are necessary for data sets that do not align to a grid and for reducing the cubic cost associated with the filtering state
>     - A structured VI approximation in which the state-space prior is only defined over the space-time points and temporal derivatives, which significantly reduces the size of the state
>     - Spatial minibatching which is necessary for large spatial data sets
>
> Each of these points is novel, and when used in conjunction alleviates many of the limitations of previous methods. For example, on our ocean currents experiment, running any competitor (AUTOIP, HELMHOLTZ, PMM) is infeasible due to the sheer size of the data set ($N=42243$).
>
> **W2: Relation to existing works.**
> Thank you for the additional literature, they are indeed related works, and we will include them in our literature review. We briefly reiterate their connection here:
>
> - Both Krämer and Hennig (2021) and Schmidt et al. (2021) are interesting works that are limited to the ODE setting.
>
> - Krämer et al. (2022): This work derives a spatio-temporal PDE solver that can be encapsulated in our framework by seeing it as an extension of the EKF prior described in Ex. 3.1. As such we do not see it as an alternative to our work but a method that can be combined with ours. This could lead to interesting directions where finite differences can be used in space whilst used in conjunction with our variational approximations enabling the application to large spatio-temporal datasets. We will add this discussion to our related work and future work section.
>
> - Duffin et al. (2021) build on the StatFEM work of Girolami et al. (2021) and Conrad et al. (2017) by deriving an extended Kalman filter where space is discretised through finite-elements. Like Krämer et al., this would be an interesting approach that could be used in conjunction with our work.

---

> > ### Comment · Reviewer_X3gM · 2024-08-09
> >
> > Thank you for the reply!
> >
> > For W1, I agree that the combination of the tools is new, even though each component may be known. My review assessed that the novelty of combining these tools may be somewhat limited, not that it doesn't exist. However, this perspective is subjective, which is why I recommend acceptance regardless.
> >
> > For W2, thank you for the clarification. I agree with your perspective on those four papers.
> >
> > Again, thank you for iterating. Since my assessed strengths and weaknesses remain, I will keep my already positive score.

---

### Author Rebuttal · Authors · 2024-08-06

We thank all five reviewers for their time and constructive reviews. This work *introduces a novel approach for solving partial differential equations (PDEs) using a Gaussian process prior* (**4ngK**) that *fits into recent literature on dynamical systems, Gaussian processes, and physics-informed machine learning* (**X3gM**), whose *beauty and originality of the proposed model is its unifying property* (**z1FQ**). We showcase our methods under a *bevy of test cases* (**BaEt**) that *convincingly demonstrate the advantages offered* (**BaEt**).

Based on the reviews, we have:
* Conducted additional experiments to showcase extensions of our proposed framework (unknown physics (**zAxE**, Q2) and extensions to AUTOIP (**z1FQ**, Q2)) and to provide additional uncertainty metrics (**BaEt**, W3).
* Improved the presentation by clarifying our problem statement (**BaEt**, W1) and our notation (see **4ngK**).
* Fixed the typos pointed out by the reviewers.

---

### Author Response · Authors · 2024-08-12

Dear Reviewers,

Thank you for the reviews and for taking the time to respond to our rebuttal. If you have any additional comments or requests we would be happy to address them in the remaining reviewer-author discussion period, and if you believe we have already addressed your corrections/questions we would appreciate it if this could be reflected in your updated evaluation.

Many Thanks,
Authors

---

### Decision · Program_Chairs · 2024-09-25

**Decision:**

Accept (poster)

**Comment:**

The paper presents a novel variational spatio-temporal Gaussian process (GP) approach for solving partial differential equations (PDEs) that efficiently handles both spatial and temporal dimensions with linear-in-time and reduced spatial complexity. This method, which incorporates physics constraints and variational techniques, outperforms existing state-of-the-art methods in both predictive accuracy and computational efficiency across various synthetic and real-world applications. Reviewers have generally praised the paper's strengths, and most concerns were addressed in the rebuttal phase. One remaining issue is readability; however, I found the paper understandable despite its technical nature. I concur with the four reviewers who recommend acceptance.

For the camera-ready version, address these two points: (1) clarify that Equation (4) represents a general form of only an evolution equation (and not a genreal PDE) and (2) note that the linear case can be handled without the need for potentials, as detailed in https://arxiv.org/abs/2212.14319.